# Cancer Vaccination and Immune-Based Approaches in Pancreatic Cancer

**DOI:** 10.3390/cancers17142356

**Published:** 2025-07-15

**Authors:** Matthew Bloom, Ali Raza Shaikh, Zhengyang Sun, Babar Bashir, Adam E. Snook

**Affiliations:** 1Department of Medical Oncology, Thomas Jefferson University, Philadelphia, PA 19107, USA; matthew.bloom@jefferson.edu (M.B.); aliraza.shaikh@jefferson.edu (A.R.S.); 2Department of Pharmacology, Physiology, & Cancer Biology, Thomas Jefferson University, Philadelphia, PA 19107, USA; zhengyang.sun@jefferson.edu; 3Departments of Medical Oncology and Pharmacology, Physiology, & Cancer Biology, Sidney Kimmel Comprehensive Cancer Center, Thomas Jefferson University, Philadelphia, PA 19107, USA; babar.bashir@jefferson.edu; 4Departments of Pharmacology, Physiology, & Cancer Biology and Microbiology & Immunology, Sidney Kimmel Comprehensive Cancer Center, Thomas Jefferson University, Philadelphia, PA 19107, USA

**Keywords:** pancreatic cancer, cancer vaccine, immunotherapy

## Abstract

Pancreatic cancer is one of the deadliest forms of cancer, with very few patients surviving in the long term even after surgery and chemotherapy. This review describes modern immune-based therapies, particularly cancer vaccines, that are being developed for the treatment of pancreatic cancer, as well as the barriers facing these efforts. Importantly, pancreatic tumors are composed of a dense layer of tissue and immune-suppressing cells that make it difficult for the immune system to attack them. Vaccines designed to train the body’s immune system to recognize and attack pancreatic cancer cells could be a promising way to improve treatment. Different types of vaccines and other immune-based therapies are described. Moreover, the causes of therapy failure so far and strategies to improve them with rational combinations are discussed. This research is crucial because it could lead to new treatments that give hope to patients with this hard-to-treat cancer.

## 1. Introduction

Pancreatic ductal adenocarcinoma (PDAC) is among the most common and lethal malignancies, with an estimated 67,440 new cases and 51,980 deaths expected in the United States (U.S.) in 2025. Despite advances in therapy, in the U.S., the 5-year survival remains poor at around 13% across all stages and only 3% for metastatic disease [1]. Surgery with pancreatectomy offers the only chance of a cure; however, only about 15–20% of patients are eligible for resection at the time of diagnosis. About an additional 30% present with borderline resectable or locally advanced disease, some of whom may undergo curative surgery following neoadjuvant therapy [2]. For those who undergo resection, adjuvant chemotherapy with a 6-month regimen of gemcitabine [3], gemcitabine plus capecitabine [4], or FOLFIRINOX (fluorouracil, leucovorin, irinotecan, and oxaliplatin) [5] has been shown to significantly improve outcomes. However, most patients still recur within 2–3 years, even with optimal multimodality treatment [3,4,5]. The role of post-operative radiotherapy has also been debated, with the most recent national trial showing a survival benefit limited to patients with node-negative disease in a subset analysis [6].

Cancer vaccines offer one potential strategy to improve outcomes in PDAC. In its simplest form, a cancer vaccine targets one or more tumor antigens to generate tumor-specific immunity that reshapes the tumor microenvironment and elicits cytotoxic T cells to eradicate cancer cells. The choice of antigen(s), delivery platform, and clinical context (primary prevention, preoperative (neoadjuvant), after resection (adjuvant), or therapeutic for advanced disease) is variable [7] (Figure 1).

Over the past 30 years, numerous vaccine candidates incorporating these elements have been evaluated in PDAC clinical trials (Table 1). This review highlights many of these studies and considers important factors influencing vaccine efficacy. These include the use of vaccine adjuvants; integration with immune checkpoint blockade, chemotherapy, or radiation; and metrics used to evaluate vaccine efficacy. A particular focus of this review is the application of cancer vaccines in resectable or borderline resectable PDAC, an area of ongoing interest to improve outcomes without the need to overcome significant immunosuppressive conditions present in advanced PDAC. This immunosuppressive milieu, which presents significant challenges to PDAC therapy, is also discussed.

In addition to cancer vaccines, many other immunotherapy approaches have been or are currently under investigation in PDAC. While immune checkpoint blockade has had limited efficacy in PDAC, emerging approaches—such as oncolytic virus therapy, adoptive cell transfer, targeting novel immune checkpoints, macrophage-directed therapies, and others—hold potential to improve outcomes and are explored in this review.

## 2. Immunologic Challenges in Pancreatic Cancer

### 2.1. Mechanistic Basis for Vaccine-Based Immunotherapy

The concept of vaccination against cancer dates to the nineteenth century, with anecdotal evidence of patients experiencing spontaneous tumor regression following acute bacterial infections. Later, Dr. William Coley treated cancer patients with extracts of heat-inactivated *Streptococcus pyogenes* and *Serratia marcescens*—a formulation known as “Coley’s toxin”—representing one of the earliest attempts at cancer immunotherapy. However, its efficacy was both debatable and unable to be reproduced at the time, resulting in harsh criticism and a lack of acceptance [44,45,46]. While Coley was ultimately vindicated when his studies were reproduced nearly a century later [47], modern cancer vaccines are typically engineered to enhance pre-existing tumor-specific T cell responses and to broaden the T cell repertoire by inducing the de novo activation of naïve or antigen-experienced T cells in an antigen-specific manner [48].

A productive and antigen-specific immune response requires four essential components from a cancer vaccine: tumor antigens, formulations, immune adjuvants, and delivery vehicles [49], many of which are discussed in more detail below. However, regardless of the vectors and components used in the vaccine, it typically requires the triggering of an inflammatory response, releasing pro-inflammatory cytokines and chemokines, and the “presentation” of tumor-associated antigens (TAAs) or tumor-specific antigens (TSAs) to T cells by antigen-presenting cells (APCs), such as dendritic cells (DCs) and macrophages, using major histocompatibility complex (MHC) molecules (Figure 2). CD8^+^ cytotoxic T cells (CTLs) recognize the presented peptides on MHC class I molecules, leading to their activation and expansion. CD4^+^ helper T cells recognize the peptides presented on MHC class II molecules and provide help signals to other immune cells, including CD8^+^ CTLs. B cells can also be activated by soluble antigens or antigens adhered to APCs, leading to the production of antibodies specific to the antigens [50]. However, the main cytotoxic effect stimulated by cancer vaccines is attributed to T cells, primarily CD8^+^ T cells.

The recognition of antigens bound to MHC molecules by T cell receptors (TCRs) on the surface of T cells is the initial step of T cell activation. To sustain proliferation and persistence, naïve T cells generally require not only signaling through the TCR (signal 1) but also costimulatory (signal 2) and cytokine (signal 3) signaling [51]. Once activated by an antigen-presenting cell, CD8^+^ cytotoxic T cells differentiate into effector cells capable of recognizing and eliminating other cells that display the same antigen [52]. Upon engaging a target cell, cytotoxic T cells can activate at least two distinct pathways to induce cell death, both of which lead to apoptosis by triggering self-destructive processes within the target cell. The first strategy utilizes perforin to assemble into transmembrane channels on the target cell surface that allow proteolytic enzymes such as granzyme B to enter the target cell cytosol. Granzyme B then cleaves and activates procaspases and triggers the apoptosis cascade. The second strategy involves the binding of Fas ligand (FasL) on the cytotoxic T cell’s surface to Fas receptor protein on the target cell, which in turn recruits procaspase-8 and initiates the proteolytic caspase cascade.

### 2.2. Suppressive Tumor Microenvironment in Pancreatic Cancer

#### 2.2.1. Overview of Tumor Microenvironment

PDAC is well known for its lethality and its low immunogenicity and immunosuppressive tumor microenvironment (TME), which together result in the limited success of immunotherapy modalities such as immune checkpoint blockade (ICBs) [53]. The TME is a complex ecosystem consisting of various cancer and non-cancer cells, an extracellular matrix (ECM), and soluble factors. Some of the major players commonly involved in the TME include endothelial cells, cancer-associated fibroblasts (CAFs), tumor-associated macrophages (TAMs), and, unique to pancreatic cancers, stellate cells [54]. Angiogenesis is one of the critical hallmarks of sustained proliferation and growth in malignancies, which is driven by the dysregulated growth of vessels formed by endothelial cells [55]. CAFs are a specialized group of fibroblasts and one of the most prominent stromal cells that are multifunctional in the TME, and they are the major producers of the extracellular matrix. Importantly, a robust desmoplastic reaction is one of the hallmarks of PDAC. Pancreatic stellate cells (PSCs) are myofibroblast-like cells within the exocrine areas of the pancreas. These cells are also critical drivers of desmoplasia and the PDAC TME and are an emerging target for therapeutics [56]. TAMs include classically activated M1 macrophages and alternatively activated M2 macrophages, with the former exerting anti-tumor functions and the latter inhibiting tumor cytotoxicity and promoting the metastasis of cancer cells [57]. Together, PDAC cells exist in an “impenetrable” network comprising immune cells, cytokines, metabolites, fibroblasts, and desmoplastic stroma rich in hyaluronan [58]. Here, we focus on the immunosuppressive aspects of TME and the challenge it brings to vaccine therapy and other immunotherapies in a PDAC-specific context.

#### 2.2.2. Immunosuppressive Immune Cells in PDAC

PDAC stands out with an imbalance in pro-tumorigenic and anti-tumorigenic immune cells in its TME [59]. Immunosuppressive cell infiltration can be detected even in the earliest lesions of PDAC development, whereas antitumor effector T cells are scarce in the tumor microenvironment [60]. Some of the signature immunosuppressive cell populations include myeloid-derived suppressor cells (MDSCs), regulatory T cells (Tregs), M2-type macrophages, stellate cells, and others that are beyond the scope of this review.

##### Myeloid-Derived Suppressor Cells (MDSCs)

MDSCs are a heterogeneous population of activated immature myeloid cells characterized by a morphological mixture of granulocytic (G-MDSCs) and monocytic cells (M-MDSCs) but lacking the expression of cell-surface markers that are specific to fully differentiated monocytes, macrophages, or dendritic cells [61]. Human G-MDSCs are HLA-DR^−^ CD33^+^ CD11b^+^ CD15^+^ CD14^−^, and M-MDSCs are HLA-DR^low/−^ CD11b^+^ CD14^+^ CD15^−^ [62]. Extensive preclinical tumor models have demonstrated the strong effects of MDSCs in masking the antitumor efficacy of immune checkpoint blockade and chemotherapies [63,64]. Notably, in most orthotopic tumor models, G-MDSCs predominate the total MDSC population (70–80% or higher), with lesser amounts of M-MDSCs (20–30%) [65]. In clinical studies, the frequency of G-MDSCs is also found to be significantly elevated in the peripheral blood of patients with pancreatic cancers [61]. Additionally, further characterization showed that myeloid cells infiltrating pancreatic tumor tissue are also predominantly the G-MDSCs subset. However, while there is disagreement about the correlation between the circulating levels of MDSCs in peripheral blood and cancer stage, circulating MDSCs may serve as a predictive marker for chemotherapy failure [61,66,67].

Treatments targeting MDSC dysregulation in combination with standard-of-care therapy have been attempted in PDAC. In one study, MDSC-targeting chemotherapy improved survival following cytokine-induced killer cell therapy [68]. Other attempts to override the upregulation of MDSCs in PDACs have also been explored in clinical trials, including adjuvant GM-CSF, an anti-inflammatory triterpenoid, and an agonistic TRAIL-R2 antibody, with improved immune responses but varied survival effects [69,70,71].

##### Regulatory T Cells (Tregs)

Tregs are classically defined as the CD4^+^ Foxp3^+^ subset of T cells that usually function as master regulators of immunological self-tolerance, which might be dysregulated in autoimmune diseases. In the context of PDACs and other cancers, they are known to suppress antitumor immunity and are thought to contribute to the failure of immune-based treatments. Several possible mechanisms of the immune-suppressive effects of Tregs have been elucidated, including reduced MHC class II antigen presentation by APCs, the cell-contact-dependent inhibition of dendritic cells, direct cytotoxicity, and the generation of inhibitory metabolites and cytokines [72,73,74,75].

In the PDAC TME, it has been observed that the prevalence of Tregs was significantly increased in PDACs compared with that in the stroma of non-neoplastic inflammation (*p* < 0.0001) [76]. The increased prevalence of Tregs was significantly correlated with clinicopathologic factors and inversely correlated with prognosis. Compared to premalignant lesions, the prevalence of Tregs was also found to be increased in malignancy. The mechanism(s) underlying the accumulation of Tregs in the PDAC TME have been extensively studied. The most likely source of tumor-associated Tregs is through recruitment from the peripheral blood or tumor-draining lymph nodes, which might later expand in the tumor microenvironment and acquire their specific patterns locally [77]. Chemokine receptors play an important role in the trafficking of Treg cells, including CCR4 and its ligand CCL22; CCR8 and its ligands CCL1, CCL8, CCL16, and CCL18; the CCR2 axis; and many others [77,78,79]. In vivo studies using mice deficient in CCR4 or CCR8 showed that Tregs were unable to infiltrate inflamed tissue and failed to control immune responses in various models, supporting their necessity in Treg accumulation in inflammatory environments [78,80]. This also supports potentially using monoclonal antibodies to target these axes and enhance immune responses [81]. PDACs might also overexpress FOXP3 (the master regulatory transcription factor driving Treg development/function), and cancer-FOXP3 was found to directly activate CCL5. In turn, CCL5 promotes the recruitment of Treg cells from peripheral blood to the tumor site [82]. On the other hand, other studies have demonstrated that KRAS G12D mutation (>90% of PDAC have mutant KRAS) modulates the expression of the cytokines IL-10 and TGFβ by tumor cells, and both convert and accumulate naïve CD4^+^ T cells into Tregs, which can be reversed by inhibiting the RAF-MEK-ERK pathway [83]. The combinational loss of RNF43 also facilitates Treg infiltration in PDACs [84]. In addition, one study found high N6-methyladenosine mRNA modification to be associated with a high density of Tregs as well [85,86].

##### Tumor-Associated Macrophages (TAMs)

Like MDSCs, preclinical studies have demonstrated an unequivocal role of macrophages in PDAC by contributing to carcinogenesis, desmoplasia, immune suppression, angiogenesis, invasion, metastasis, and drug resistance [60,87,88]. They are one of the earliest infiltrating cells in PDACs and continue to increase during progression to invasive cancer, originating from circulating monocytes and being recruited by chemotactic signals [89,90]. Similarly to other macrophages, TAMs can adapt their functional characteristics—termed polarization—in response to environmental cues, which include stimuli derived from pathogens, stromal and immune cells, and the extracellular matrix and metabolites [91]. As a result, they are commonly divided into two main subtypes, “classically activated” M1 or “alternatively activated” M2 [92].

The immune-suppressive aspects of TAMs are mainly associated with their polarization to M2 macrophages, which normally play a pivotal role in wound healing and the suppression/regulation of inflammatory responses. In PDAC, the immunosuppressive effects of these protumor TAMs arise from their ability to inhibit the antitumor activity of CD8^+^ cytotoxic T lymphocytes by upregulating PD-L1 expression in tumor cells and depleting essential nutrients required by T cells [93]. It was also discovered that TAMs suppress adaptive immune responses via the Dectin-1/Galectin-9 axis and promote the production of immunosuppressive factors, such as CXCL1 and CXCL5, in tumor cells through the upregulated expression of apolipoprotein E (ApoE) [94,95,96]. Beyond their role in immune suppression, TAMs support cancer cell survival by secreting growth factors such as TGFβ and by producing cytokines and chemokines that promote tumor metastasis through both direct and indirect mechanisms.

##### Pancreatic Stellate Cells (PSCs)

Pancreatic stellate cells (PSCs) were initially identified as the primary culprits underlying fibrogenesis in pancreatitis and were later found to strongly support tumor growth [97,98,99]. Under normal conditions, PSCs are quiescent as lipid storage cells. However, when under chronic inflammation, oxidative stress, or hypoxic conditions, PSCs can be activated, transdifferentiate into highly proliferative myofibroblasts, and lead to the deposition of collagens and the production of the extracellular matrix (ECM), which induces the desmoplastic transformation of the TME, a hallmark of PDAC [100,101,102,103].

The activation of quiescent PSCs is driven by a combination of cytokines, including IL-1, IL-6, IL-10, and IL-17, which are highly correlated with poor survival outcomes in PDAC [104]. After activation, PSCs primarily support the proliferation and metastasis of PDAC by mechanically reprogramming the tumor stroma, providing metabolic advantages, enhancing chemoresistance, promoting angiogenesis, and sequestering immune cells. In the context of PDAC vaccines, activated stellate cells create a physical barrier that isolates immune cells from directly contacting PDAC cells, reprogram the TME metabolism, and manipulate crosstalk between PSCs and immune cells through cytokine signaling [102,105].

In mouse models, PSCs secrete FGF-2, TGFβ, and PDGF [97], which lead to the establishment of intense fibronectin and collagen deposition around cancer cells, limiting T cell infiltration and function [106]. Additionally, these rigid ECM components induce the Rho-associated coiled-coil kinase-dependent activation of FAK1, a tyrosine kinase that regulates T cell survival, antigen sensitivity, cytokine production and migration, contributing to suppressed anticancer immunity [102,107]. This desmoplastic response also creates hypoxic and avascular conditions, which impose considerable metabolic constraints on immune cells. During metabolic reprogramming, while cancer cells can express more nutrients import molecules (e.g., GLUT1, MCTs, ASCT2, LATs) to obtain fuel sources for survival, effector T cell proliferation and activation are restricted due to the crucial lack of tryptophan and arginine, caused by elevated indoleamine-2,3 dioxygenase 1 (IDO1) and arginase (ARG1, ARG2) in metabolically altered stromal cells. Moreover, the metabolic competition, including a lack of glucose, impairs IFNγ secretion and the function of T cells [108].

In the context of cytokine and chemokine signaling, activated PSCs can secrete SDF-1α/CXCL12 to promote CD8^+^ T-cell chemotaxis toward the juxtatumoral stroma, thereby preventing CD8^+^ T cells from accessing tumor nests [109,110]. PSCs also produce numerous other soluble cytokines, including IL-6, M-CSF, and VEGF, that promote the differentiation of myeloid cells into MDSCs [109]. Conversely, IL-6 or SDF-1α/CXCL12 secreted by M2 macrophages can also promote CAF/PSC activation and desmoplasia, creating a positive-feedback cycle driving desmoplasia and the suppression of antitumor immunity [109,111]. Blocking the CXCL12/CXCR4 axis can promote T-cell accumulation and activity, which can synergize with PD-L1 blockade [112]. Activated PSCs also directly crosstalk with NK cells and suppress their cytotoxic functions, as established in co-cultured models. However, the exact mechanism remains unillustrated [113].

#### 2.2.3. Lack of Effector T Cell Infiltration and Antigen Recognition

PDAC is classified as an immunologically “cold” tumor, characterized by the poor infiltration and support of effector T cells. Therefore, the majority of PDAC patients do not benefit from immune checkpoint blockade (ICB). Moreover, human PDACs typically harbor dysfunctional T cells within the tumor microenvironment. Even in early-stage, resectable cases, functional T cells are scarce and often exhibit phenotypes of exhaustion, indicating prior antigen exposure but impaired effector function [114]. The exact mechanism behind this phenomenon is unclear, with multifaceted influence from the physical barrier created by tumor stroma and the extracellular matrix, lack of antigen presentation, and regulatory immune cells and cytokines, as mentioned above [115]. However, controversies exist as to whether the T cells (1) were initially attracted to and activated by premalignant or early malignant cells but later become exhausted or tolerant or (2) were never alerted to the developing and progressing malignancy. However, there has been increasing evidence in preclinical and clinical studies showing that in the earliest premalignant lesion, pancreatic intraepithelial neoplasia-1 (PanIN 1), T cells are initially attracted by KRAS mutations, and they later decline as tumors transform into PDAC [116,117].

The dysfunction of CTLs also results from inadequate antigen presentation to T cells in the PDAC TME. In human pancreatic carcinoma tissues, a reduction in or loss of HLA class I and TAP (both necessary for antigen presentation) was observed in 76% of samples, while the loss or downregulation of TAP was observed in 53% [118]. The reduction in class I and TAP expression was reversible upon exposure to IFNγ in vitro, suggesting a secondary regulatory mechanism. Various factors have been attributed to the deficit in antigen presentation in PDAC, including focal adhesion kinase (FAK) signaling, the autophagy pathway, MAPK signaling, low tumor mutational burden, and other tumor intrinsic factors [119,120,121,122,123]. Moreover, driver KRAS mutations (mKRAS) present in >90% of PDAC [124] are associated with over-expression of CXCL5, a known recruiter of TAMs, TANs, and MDSCs, which suppress effector T cells [96,125,126]. mKRAS expression is also associated with PD-L1 upregulation [127]. Further, aberrant Wnt/β-catenin signaling is also common in PDAC due to recurrent mutations in negative regulators such as Rnf43 [84,128]. The nuclear translocation of β-catenin results in the transcriptional upregulation of numerous genes that underlie the hallmarks of cancer, such as cell cycle progression, epithelial–mesenchymal transition (EMT), and angiogenesis. These genes include cyclin D1, cyclin E, MMP-7, c-myc, VEGF, and many others. Wnt signaling also promotes resistance to apoptosis and maintenance of cancer stem cells, leading to therapeutic resistance [129]. Taken together, tumor intrinsic factors that induce immune suppression, combined with a lack of adequate antigen presentation and T cell infiltration, create challenges to generating a strong and enduring cytotoxic immune response in PDAC.

#### 2.2.4. Stromal Cells and Extracellular Matrix

The PDAC TME is characterized by extensive desmoplasia with predominant cancer-associated fibroblasts. One subset of fibroblasts (myofibroblasts), together with tumor cells and macrophages, secrete various proteins into the surrounding tissues, forming an extracellular matrix (ECM) with complex properties. The ECM consists of a dense, non-cellular mixture of proteins, glycoproteins, proteoglycans, and polysaccharides and plays an important role in pancreatic cancer progression [60,94]. The ECM can be divided into the basement membrane (BM) that supports epithelial/endothelial cell behavior and the interstitial matrix (IM) that supports the underlying stromal compartment [130]. While fibrillar-forming collagens type I, II, III, V, XI, XXIV, and XXVII, as well as the beaded filament-type VI collagen produced by the fibroblasts in the stroma, make up most of the interstitial matrix, the network-forming collagens, such as type IV and type VIII collagen, are the main constituents of the basement membrane. Collagens are often crosslinked and linearized, leading to the increased stiffening of the PDAC tissue, which elicits various effects on the cancer cells, including cell differentiation, proliferation, differential gene expression, migration, invasion, metastasis, and survival. Moreover, highly aligned stromal collagen has been identified as a strong negative prognostic factor following pancreatic ductal adenocarcinoma resection [131].

Beyond producing a dense extracellular matrix, CAFs exert three major functions in the TME: responding to tumor signaling, restricting tumor-infiltrated immune cells, and reprogramming tumor metabolism. CAFs also interact with immune cells and suppress antitumor immunity by secreting GM-CSF, M-CSF, IL-6, and CXCL12 [132,133]. In PDAC mouse models, depleting CAFs that express fibroblast activation protein (FAP) not only inhibits tumor growth alone but also potentiates the efficacy of anti-CTLA-4 and anti-PD-L1 ICB antibodies [112]. Interestingly, while the depletion of FAP+ CAFs results in increased animal survival, the depletion of CAFs expressing alpha smooth muscle actin (αSMA) leads to increased tumor growth and decreased animal survival. Together, this suggests that the functional heterogeneity of CAFs may play a role in PDAC progression and response to therapy [134].

Recently, the presence and significance of the tumor microbiome (the microbiome within tumors) in PDAC stroma has been increasingly emphasized [135]. Pushalkar et al. identified a distinct stage-specific gut and pancreatic microbiome signature that is associated with PDAC disease progression by inducing intra-tumoral immune suppression. Conversely, the ablation of the microbiome markedly protected against PDAC and enhanced antitumor immunity and efficacy of immunotherapy. Interestingly, even the transfer of bacteria from PDAC-bearing hosts, but not controls, results in accelerated tumor growth in recipients. Other studies have shown that the fungal microbiome in PDAC drives IL-33 secretion, which in turn results in increased Th2 infiltration and tumor growth [136]. Another study found that the metabolism of tryptophan by *Lactobacillus* resulted in the increased activity of Aryl hydrocarbon receptor (AhR) on TAMs, and the deletion or pharmacologic inhibition of AhR in myeloid cells decreased PDAC growth, improved the efficacy of immune checkpoint blockade, and increased the intra-tumoral frequencies of IFNγ^+^CD8^+^ T cells [137]. These studies highlight a critical but currently understudied intersection between immune surveillance, gut and tumor microbiome, metabolism in the TME, and tumor cell growth.

### 2.3. Strategies to Overcome PDAC Immunological Challenges

Despite the immunosuppressive nature of the PDAC tumor microenvironment, vaccines remain a promising therapeutic approach to overcome the issue of the low immune recognition of PDACs. By enhancing antigen presentation and driving the expansion of tumor-specific T-cell clones, PDAC vaccines aim to elicit de novo or boost pre-existing immune responses, and importantly, they can be easily combined with other therapeutic adjuvants that directly target the TME [138]. These include the blockade of CXCR2, CCR2, and CCR5, which blocks the increased infiltration of M2 macrophages and MDSCs in PDAC [139]. Chemotherapies and monoclonal antibodies are also often used to ablate Tregs in preclinical models [140,141] and are being revisited in the clinic. Targeting the tumor stroma, FAK inhibitors have also been tested for the reduction of fibrosis and increased tumor cell antigen presentation [107,119]. Ruxolitinib, a small-molecule JAK1/JAK2 inhibitor, has been tested in a phase 2 clinical trial to prevent the activation of pancreatic stellate cells and the induction of inflammatory CAFs. While there was no observed benefit in OS, a significant increase in OS was noted in patients with systemic inflammation compared with the placebo plus capecitabine group [142,143].

Extensive efforts have been spent to identify not only the potential targetable neoantigens in PDACs but also the qualities of these antigens, which are significantly correlated with eliciting robust CD8^+^ T cell responses and patient survival [144,145]. It is believed that by successfully delivering high-quality neoantigens and/or self-antigens and reinvigorating antitumor immunity, a new generation of cancer vaccines will alter the landscape of PDAC treatment paradigms and improve patient outcomes [122].

## 3. Considerations for Cancer Vaccines in PDAC

### 3.1. Vaccination in the Neoadjuvant and Adjuvant Setting

Vaccine studies have been explored at various time points throughout the treatment and surveillance paradigm of resectable, borderline resectable, and locally advanced pancreatic cancer to improve current standards of care. In this context, vaccines are administered in the neoadjuvant or adjuvant settings, often combined with chemotherapy or radiotherapy. In the neoadjuvant setting, vaccines have been explored to improve conversion rates to surgery [25], while in the adjuvant setting, treatment aims to improve cure rates or delay recurrence [11,31].

An active area of investigation has been in improving outcomes for patients with a high risk of recurrence following standard therapies. This may be an opportune time for vaccination, as it can prevent recurrence by targeting micrometastatic disease [10,11,26]. In contrast, in advanced pancreatic cancer, disease progression is often inevitable, and vaccine trials in PDAC have generally shown little success [14,16,20,36,40,43]. This lack of efficacy may be due to local immunosuppressive effects, rapid tumor growth, or the insufficient expansion of CTLs. Additionally, higher tumor burdens have been associated with reduced effectiveness of immunotherapy, with higher levels of T cell senescence and exhaustion [146].

### 3.2. Combination with Chemotherapy or Radiation

Cytotoxic therapies, including chemotherapy and radiotherapy, not only eradicate tumor cells but also influence the tumor immune microenvironment. This immunomodulatory effect can be harnessed with vaccines to enhance their effectiveness. While these therapies may stimulate immune responses, they may also suppress immune activation. Therefore, carefully optimizing combination approaches is essential to induce a proinflammatory, antitumor immune response while mitigating immunosuppression.

There are several ways in which cytotoxic therapies have been shown to favorably shape the antitumor immune response. They can promote leukocyte infiltration into tumors [147,148], reduce immunosuppressive cell populations [149,150], stimulate chemokine release [148,151], reduce HLA defects [147], remodel tumor stroma [150], and induce immunogenic cell death, enhancing immune system recognition [152].

In pancreatic cancer, gemcitabine has been shown to activate naïve T cells [153] and increase CD11c^+^ DCs and CD14^+^ monocyte populations [154]. Given its effects on DCs, gemcitabine has been explored for its potential role in enhancing DC-based vaccine strategies [23,29,155]. 5-FU increases cytokine production and levels of natural killer (NK) cells and lymphokine-activated killer (LAK) cells both in vitro and in vivo [151]. FOLFIRINOX is associated with a higher CD8–CD4 T-cell ratio, the shifting of macrophages towards an M1-polarized phenotype, and the reduced density of CD15^+^ARG1^+^ immunosuppressive granulocytes [149]. Additionally, PDAC patients treated with FOLFIRINOX have higher CD8^+^ T-cell infiltration and lower Treg cell and M2 macrophage density in the TME [147]. Radiotherapy, through IFNγ-driven changes, enhances T-cell recruitment and tumor antigen presentation by increasing MHC class I expression on tumor cells. IFNγ also increases VCAM-1 expression and induces several chemokines, promoting immune infiltration [148].

Alongside immune-stimulatory signals, cytotoxic therapies can also promote immunosuppression, which can occur together to form a complex immune environment. For example, while gemcitabine may activate naïve T cells, it can simultaneously lead to a decline in memory T cells [153] and promote immunosuppressive effects by increasing the synthesis of chemokines and TGFβ, leading to changes in the tumor stroma [156]. Recent data suggests that tumor immune profiles in PDAC in response to chemotherapy may even differ by sex, with women having a more favorable immune environment that hinders the infiltration of M2 macrophages into the TME [157].

Over the years, numerous studies in resected pancreatic cancer patients have explored the combination of adjuvant chemotherapy and/or chemoradiotherapy with vaccination. Some of these trials involved chemotherapy with gemcitabine [9,10], which was previously considered the standard of care when these trials were undertaken. Other trials in resected PDAC patients have examined combinations such as S-1 with or without gemcitabine [23], chemoradiation [13,32], SBRT and mFOLFIRINOX [33], atezolizumab and mFOLFIRINOX [39], and gemcitabine and chemoradiation [37]. In a trial of personalized neoantigen mRNA vaccination administered alongside atezolizumab and FOLFIRINOX, neoantigen-specific T cells persisted up to 2 years, suggesting that durability was not affected by post-vaccination chemotherapy [39]. In the neoadjuvant setting, vaccination has been tested alongside FOLFIRINOX or gemcitabine/nab-paclitaxel and chemoradiation in a phase III trial, but no improvements were observed in PFS or OS [38].

The optimal timing of vaccination in relation to cytotoxic therapy is uncertain. In the adjuvant setting, clinical trials have explored administering vaccines both before [31,39] and after [13,25] standard-of-care treatment, as well as sequential [25,39] or concurrent approaches [9,10]. The TeloVac study investigated both strategies—administering the vaccine either alongside or following chemotherapy—but found no improvement in OS with either approach [14]. It is also important to consider minimizing delays in standard-of-care cytotoxic therapy, particularly in the adjuvant setting, where initiation within 8–12 weeks of surgery, following adequate recovery, has become a benchmark for phase III trials [158]. However, short delays may be acceptable if allowing for an optimized immune response without significantly compromising outcomes.

### 3.3. Combination with PD-1/CTLA-4 Blockade

Immune checkpoint blocking (ICB) therapies have become well-established for various cancers. Through several not yet well-defined mechanisms, including the blockade of inhibitory signals acting on effector T cells, ICB can induce immune-mediated eradication of tumors. In pancreatic cancer, however, their use has been limited to patients with microsatellite instability-high tumors (resulting in high neoantigen burdens), and even in this group, objective responses occur in a subset of patients [159]. Nevertheless, they remain an attractive option for investigation in combination with vaccines to bolster immune responses.

Several completed trials have explored combination approaches using vaccines and immune checkpoint blockade [35,39,160]. Ipilimumab, an anti-CTLA4 antibody, was tested with or without GVAX vaccination in patients with advanced disease. Ipilimumab was administered at a relatively high dose of 10 mg/kg every 3 weeks for a total of four doses, followed by maintenance dosing every 12 weeks. In the arm with both GVAX and ipilimumab, 3 out of 15 patients had prolonged disease stabilization (31 to 81 weeks), and 7 out of 15 had improvements in CA19-9 levels. The 1-year OS in the GVAX and ipilimumab arm was 27% compared to 7% in the ipilimumab arm. Grade 3–4 immune-related adverse events were observed in 20% of patients in each arm [35]. Thus, combining vaccination with ipilimumab has potential benefit, and it is being explored in ongoing clinical trials.

Another study in advanced disease evaluated GVAX and CRS-207 with or without nivolumab, an anti-PD1 antibody. Nivolumab at a dose of 3 mg/kg was administered on the day prior to GVAX or CRS-207 every 3 weeks for six cycles. There was no difference in OS in the two groups, although the disease control rate, 12- and 18-month OS rate, and CA19-9 dynamics all favored the nivolumab arm. Patients who received nivolumab and had a longer OS showed increased CD8^+^ T-cell densities and decreased myeloid cell densities in tumor samples [160].

In a separate analysis comparing this study with another trial of GVAX plus ipilimumab, differences in peripheral blood immune responses were observed between the two ICB antibodies. Both studies demonstrated a diversification in T-cell repertoires, but the changes were more significant in patients receiving ipilimumab. Additionally, in patients treated with ipilimumab, a higher degree of T-cell clonal expansion was associated with a longer OS [161]. The study may support a strategy of using anti-CTLA4 to initiate a robust T-cell response, followed by anti-PD1 to promote T-cell expansion and maintenance.

### 3.4. Adjuvants

Adjuvants are substances added to vaccines to increase immunogenicity. They have been well studied in infectious disease vaccines and are also key components of cancer vaccines. Adjuvants can be categorized into two main types: delivery systems, such as lipid particles or microparticles, which are carriers that can enhance local immune responses, and immune potentiators, which mimic natural infections to activate pattern-recognition receptors (PRRs) on immune cells [162]. Activated PRRs can then trigger the activation and maturation of APCs and the release of cytokines and chemokines, which stimulate adaptive immune responses and further strengthen the overall immune response [163].

A variety of adjuvants have been used in pancreatic cancer vaccine trials in various contexts. GM-CSF, also a hematopoietic factor, is commonly used due to its ability to enhance local antigen processing and presentation and promote effector T-cell activation [162]. However, its effectiveness as an adjuvant has been questioned, as it may increase MDSCs [164,165], although this was not observed in a PDAC vaccine clinical trial when the vaccine with GM-CSF combined with chemotherapy was compared to chemotherapy alone [14]. In PDAC vaccine trials, GM-CSF is typically given as one or multiple intradermal or subcutaneous injections around the time of vaccination [8,9,14,15,42]. GM-CSF has also been used in whole-cell vaccines, where pancreatic cell lines are engineered to secrete GM-CSF (GVAX) to recruit and provide maturation signals to DCs at the vaccine site [31,32,33,34,35,36]. Many patients experience mild local toxicities at the vaccination site due to local immune responses from GM-CSF secretion. A systemic rise in GM-CSF can also be detected up to 96 h after vaccination, with peak levels at around 48 h [31].

In addition to GM-CSF, various adjuvants have been used in pancreatic cancer vaccine trials. These include cytokine-based adjuvants (IL-12 [166]), toll-like receptor agonists (CpG-7909 [11] and poly-ICLC [30]), bacterial-based adjuvants (BCG [12] and OK-432 [23]), and delivery-based adjuvants (Freund’s Adjuvant [19] and SB-AS2 [13]). In a PDAC vaccine trial comparing vaccination with or without an IL-12 adjuvant, the addition of IL-12 led to increased IFNγ production, activation of CD4^+^ and CD8^+^ T cells, and enhanced CD8^+^ CTLs [166]. Another study with CpG-7909 combined with a KRAS vaccine for PDAC tested escalating doses of the adjuvant, with the highest dose being well tolerated and all patients demonstrating a vaccine response in this cohort [11]. As adjuvants continue to be incorporated into future trials, their ability to enhance immune responses without introducing unacceptable toxicity should continue to be followed.

### 3.5. Priming Doses, Boosters, and Heterologous Prime-Boost Vaccines

Vaccine boosters are administered to maintain tumor-specific immune responses over time. The timing of boosters in cancer vaccine clinical trials is influenced by various factors, including preclinical data, extrapolation from similar trials, the type of vaccine (e.g., peptide, DC-based, whole tumor cell), and the concurrent use of adjuvants or immune modulators, such as with ICB.

Some PDAC vaccination trials implement multiple initial priming doses given over a short period, followed by booster doses. In one personalized neoantigen mRNA vaccine trial, after a dose of atezolizumab, patients received seven weekly priming doses, an additional priming dose, and a booster dose after adjuvant chemotherapy. This schedule proved effective, with vaccine-expanded T cells persisting for up to two years, even after post-vaccination chemotherapy. In addition, the booster dose successfully re-expanded primed T-cell clones in responders [39].

Heterologous prime-boost vaccination is another strategy used to enhance vaccine efficacy. In this approach, an initial vaccine is administered to “prime” the immune system, followed by the use of a different vaccine platform (“boost”) to enhance the immune response. This strategy was tested in a PDAC trial of GVAX (GM-CSF-expressing PDAC cell lines) and CRS-207 (mesothelin-expressing *Listeria monocytogenes*). This strategy could induce qualitative changes in the immune response with an intracellular bacterium, expand mesothelin-specific T-cell responses, and possibly reshape the TME due to *Listeria* infiltration into tumors. In the experimental arm of the trial, patients received two doses of GVAX, followed by four doses of CRS-207. Although well tolerated, there was no improvement in OS compared to chemotherapy in the phase IIb study [36].

Another heterologous prime-boost study used a vaccinia virus vector, followed by a fowlpox virus vector, to deliver CEA and MUC-1 tumor antigens. The use of a fowlpox vector for boosting could allow the boost to evade vaccinia-specific neutralizing immunity induced during priming. In the phase I study, 5 out of 10 patients developed increased anti-CEA antibody titers. As can be observed with viral vectors, all patients in the study developed antibodies to the vaccinia virus, with high titers possibly impairing the induction of MUC-1- or CEA-specific T cells, although this relationship was not definitively confirmed due to the low number of patients [42].

### 3.6. Outcome Measures

Cancer vaccines and other cancer therapies aim to improve clinical outcomes, such as prolonging patient survival, promoting tumor shrinkage, or, in the adjuvant setting, improving RFS. Early-phase vaccine trials evaluate safety and immunogenicity, which may suggest clinical relevance, although immunological endpoints do not always correlate with efficacy [167]. Unfortunately, no single correlative assay can reliably predict a productive antitumor response. However, trials have used a variety of immunological biomarkers, such as the expansion of antigen-specific T cells, cytokine production, and other immune markers, for immunological monitoring. Additionally, alongside traditional tumor markers such as CA19-9, circulating tumor DNA (ctDNA) is emerging as a tool for assessing treatment response, particularly in the adjuvant setting.

#### 3.6.1. Immunological Outcomes

A range of immunological responses are assessed in PDAC vaccine trials using various assays. One method is the measurement of a delayed-type hypersensitivity (DTH) reaction after vaccination. This test assesses the T-cell response to a particular antigen or multiple antigens following their administration, typically intradermally [8,9,12,13,14,15,25,29,168]. This is similar to the PPD skin test for detecting an immune response to tuberculosis in infected people. Whole tumor cells, rather than purified antigens, may also be used as a stimulus [31]. Studies in melanoma patients have shown a correlation between DTH reactivity and clinical outcomes [8,169,170]. Biopsies of skin sites may also be assessed for the histological analysis of T-cell infiltration [12].

A variety of ex vivo and in vitro methods can also be used to assess antigen-specific T-cell responses. These assays may measure IFNγ or granzyme release from antigen-specific T cells [11,13,20,22,23,25,29,42,166], MHC class I tetramer analysis [30], or T-cell receptor diversity [11,39]. Changes to other cell types, such as MDSCs and Tregs, are also commonly measured [10,25,41], as are antibody responses [13,16,17,18,42].

As described below, vaccination with GVAX induces the development of tertiary lymphoid aggregates within the TME of resected tumors [32,34]. These structures resemble ectopic lymph nodes, consisting of organized clusters of immune cells, and have been observed in other malignancies as well [171]. Their presence is correlated with better clinical outcomes in malignancy [171], and in GVAX-treated patients, they are associated with prolonged survival in some cases [32]. However, their exact role in response to treatment is not completely understood and is still under investigation [171].

#### 3.6.2. ctDNA and CA19-9 Dynamics

Circulating tumor DNA (ctDNA) is an increasingly important biomarker in several cancers, including gastrointestinal malignancies. The most substantial real-world evidence for its use comes from colorectal cancer, where trials such as GALAXY, BESPOKE, and DYNAMIC have demonstrated its prognostic significance in the adjuvant or post-curative intent setting [172,173]. Notably, the clearance of ctDNA with adjuvant chemotherapy at 3- and 6-months after surgery is associated with better disease-free survival (DFS) [174]. Conversely, persistent ctDNA positivity following locoregional treatment identifies high-risk individuals who may benefit from vaccines targeting residual micrometastatic disease and improving DFS. In addition to identifying patients at high risk for recurrent disease, ctDNA clearance can potentially be used as a surrogate endpoint in PDAC cancer vaccine trials to accelerate research in this setting.

A recent review by Jonnalagadda et al. summarized studies of ctDNA in PDAC, highlighting its prognostic potential, including in the adjuvant setting [175]. One study suggests that after curative resection for PDAC, combining ctDNA with CA19-9 may improve the identification of patients at high risk for poor outcomes [176].

In a recent adjuvant trial of a KRAS vaccine in patients with PDAC and colorectal cancer, patients were enrolled based on ctDNA positivity and/or elevated CA19-9 or CEA, with no evidence of disease on imaging. A decrease in ctDNA and/or tumor markers occurred in 80% of PDAC patients. Notably, T-cell responses correlated with reductions in ctDNA and/or tumor markers, as well as delayed tumor recurrence. Additionally, although ctDNA and CA19-9 responses occurred at similar rates, ctDNA showed a greater magnitude of reduction, indicating differences that should be studied further [11].

## 4. Overview of Cancer Vaccines Used in PDAC

### 4.1. Peptide-Based Vaccines

A peptide vaccine uses short amino acid sequences derived from a tumor antigen to stimulate an immune response. These peptides can derive from TAAs or TSAs. TAAs are present in normal tissues but are often overexpressed in tumors; examples include mucin-1 (MUC-1) and mesothelin. TSAs, in contrast, are exclusively present on tumor cells, with examples such as mutated KRAS (mKRAS).

Over the past 30 years, numerous peptide-based vaccines have been evaluated in PDAC, with largely modest clinical outcomes. To improve efficacy, combination strategies incorporating adjuvants and/or a variety of other immunomodulatory agents, such as ICBs, have been explored. Another strategy involves the use of off-the-shelf multi-peptide cocktail vaccines, which can target multiple TAAs, potentially helping to address tumor heterogeneity and reduce tumor escape through antigen loss [177]. A more recent strategy has involved personalized neoantigen vaccines, which are tailored to each patient by targeting multiple peptides unique to a patient’s individual tumor [39].

In addition to the notable peptide targets listed below, peptide vaccines for PDAC have also been tested against other antigens, including VEGF receptor 2 (VEGFR-2) [41,178,179], KIF20-A [180], CEA [181], WT-1 [182,183], and chaperones of tumor-specific peptides, such as heat shock proteins (HSPs) [184].

#### 4.1.1. KRAS Vaccines

KRAS is a mutated, oncogenic driver in more than 90% of PDAC, making it a highly attractive vaccine target. The KRAS protein is a guanosine triphosphatase (GTPase) that acts as a molecular switch, cycling between an active (GTP-bound) and inactive (GDP-bound) state and regulating cell growth and survival signaling. Point mutations in KRAS can lock the protein in a constitutively active state, resulting in uncontrolled proliferation and tumor progression [185]. Overall, studies using KRAS-targeted vaccines suggest that patients who develop immune responses to the vaccine may experience a more favorable clinical course, although the results have been modest.

In a promising early study, subjects with resected (*n* = 10) or advanced (*n* = 38) PDAC were administered a mixture of four mutant RAS peptides or a single peptide matching the patient’s KRAS mutation. Before peptide injections, subjects also received a GM-CSF adjuvant. About half of the patients in the study elicited an immune response, and in those with advanced disease, measurable DTH activity and/or T-cell responses were associated with prolonged OS [8]. Several pooled mKRAS vaccine trials have subsequently followed, each incorporating a unique combination of mKRAS peptides. TG01, containing the seven most common KRAS codon 12 and 13 oncogenic mutations, showed high levels of immune activation (>90% of patients with positive immune responses) [9]. Similarly, GI-4000, a vaccine targeting the seven most common RAS mutations at codons 12 and 61, showed a subset of immune responders, although clinically, the median RFS and OS were similar to those of the placebo [10].

A key and unique study of a KRAS-targeted vaccine is AMPLIFY-201, a phase I trial of ELI-002 2P in patients with minimal residual disease (MRD) after post-surgical adjuvant treatment for KRAS-mutated PDAC and other solid tumors. The vaccine is amphiphilic, consisting of a lipid tail attached to mutated KRAS G12D and G12R peptides. This amphiphilic modification enhances peptide binding to albumin in circulation through a fatty-acid binding pocket, facilitating transport to lymph nodes for uptake by resident antigen-presenting cells (APCs). This process is akin to the identification of sentinel lymph nodes in cancer surgeries, where dyes binding to albumin are directed to draining lymph nodes. The vaccine also includes an amphiphile-modified version of a CpG oligonucleotide adjuvant, Amph-CpG-7909. This approach to using amphiphilic vaccines and adjuvants, termed “albumin hitchhiking,” has been shown to significantly increase vaccine accumulation in draining lymph nodes and subsequent T-cell priming [186].

A total of 25 patients were treated with ELI-002 2P and escalating doses of Amph-CpG-7909. MRD was determined by circulating tumor DNA (ctDNA) positivity, elevated tumor markers, or a combination of both. The trial showed high levels of immune response, with KRAS-specific T-cell responses and biomarker responses each occurring in 84% of patients (21 of 25). T-cell responses above the median were correlated with biomarker response. Notably, biomarker clearance also occurred in 24% of patients (6 out of 25), including 3 patients with pancreatic malignancy [11]. Attempting to build upon these results, the AMPLIFY-7P trial is currently evaluating ELI-002 7P, which contains seven KRAS and NRAS peptides, along with the previous recommended phase 2 dose (RPTD) of Amph-CpG-7909 in patients with KRAS-/NRAS-mutated PDAC and other solid tumors [187].

#### 4.1.2. Mucin-1 Vaccines

Tumor-associated mucins, including Mucin-1 (MUC1), have long been studied as potential TAAs in pancreatic cancer and other epithelial tumors [188]. In malignancy, MUC1 undergoes hypoglycosylation and loses its polarization, thereby promoting CTL responses and low levels of MUC-1-specific antibodies [189]. The use of peptide vaccinations targeting MUC-1 in PDAC has been investigated but is limited.

In a phase I trial of patients with PDAC and other adenocarcinomas who received a synthetic MUC1 peptide vaccine and Bacillus Calmette–Guérin (BCG) adjuvant, 7 of 22 patients showed a 2- to 4-fold increase in mucin-specific CTLs in the peripheral blood. The vaccine was generally well-tolerated, with most patients experiencing local ulceration at the injection site [12].

In another phase I study, patients with resected or locally advanced pancreatic cancer were given a synthetic MUC1 peptide vaccine with SB-AS2 adjuvant, followed by adjuvant therapy at the physician’s discretion. One goal of the study was to effectively induce MUC1-specific helper T-cell responses. Prior research has suggested that the weak natural immune response to MUC-1 may result from the inefficient processing of tumor MUC1 protein by APCs, leading to the insufficient activation of MUC-1-specific helper T cells [190]. Based on this, a MUC-1 synthetic peptide vaccine, shown to induce helper T-cell responses in preclinical studies [191], was tested in this trial. A total of 16 subjects received one of four escalating doses of the vaccine in cohorts of 4 patients. The vaccine was well tolerated with common adverse events (AEs) including grade 1 flu-like symptoms and injection site reactions. In 5 of 16 patients, MUC-1 specific IgG antibodies were produced, which is an indirect measure of potential peptide-specific helper T-cell activation, promoting class switching from anti-MUC1 IgM. Overall, the vaccine induced low but measurable mucin-specific immune responses in a subset of patients [13].

#### 4.1.3. Telomerase Vaccines

Telomerase, a reverse transcriptase enzyme, has long been recognized for its ability to maintain the ends of eukaryotic chromosomes, known as telomeres [192]. By counteracting telomere shortening, telomerase can extend the number of cell divisions of cancer cells, preventing senescence. Telomerase activity has been detected in up to 95% of pancreatic tumors [193]. One of the components of telomerase, TERT, has gained attention as an antigen of interest for immune cell targeting with vaccination [194]. While telomerase is highly overexpressed in malignant cells, its presence in some normal tissues, such as stem cells and epithelial cells in colonic crypts, makes monitoring for on-target toxicities of TERT-targeted vaccines particularly important [195].

GV1001, a synthetic peptide corresponding to the human TERT (hTERT) fragment, has been tested in multiple trials in PDAC patients. In a phase I/II dose-escalation study, patients with advanced pancreatic cancer were vaccinated with GV1001, along with adjuvant GM-CSF. Vaccine-induced immune responses occurred in 63% of patients, with the highest response rate of 75% observed in those receiving the intermediate dose. Patients in the intermediate dose group also had a significantly longer survival of 8.6 months compared to other groups (intermediate vs. low: *p* = 0.006; intermediate vs. high: *p* = 0.005) [15].

In a subsequent phase III trial, the TeloVac study, GV1001 with GM-CSF adjuvant, failed to show an improvement in OS when added to chemotherapy alone in patients with advanced pancreatic cancer. The vaccine was tested both sequentially and concurrently with chemotherapy [14]. While cytotoxic chemotherapy is traditionally viewed as immunosuppressive due to its destruction of rapidly dividing cells, including immune cells, paradoxical evidence suggests that it may enhance T-cell-mediated antitumor immunity. This could occur through several mechanisms, including increased tumor antigen cross-presentation, enhanced T-cell expansion, and increased tumor infiltration by T cells [196]. Although 38% and 37% of patients in the TeloVac study had immune responses to sequential immunotherapy and concurrent chemoimmunotherapy, respectively, this did not translate to improved clinical efficacy in terms of survival [14].

Other telomerase-targeting vaccines tested in pancreatic cancer are INO-1400 and INO-1401, DNA vaccines with plasmid delivery encoding one of two modified versions of full-length hTERT. In a phase I trial, INO-1400 or INO-1401 was administered with or without plasmid DNA encoding IL-12 as an adjuvant. The vaccine was well-tolerated, and 96% of patients across various malignancies had de novo cellular immune responses or enhanced pre-existing cellular responses to native hTERT. In pancreatic cancer patients who received the vaccine following surgery and standard therapy, hTERT-specific cytotoxic T-cell responses were associated with improved DFS. The study also showed that the addition of IL-12 led to a higher magnitude of cellular responses [166].

#### 4.1.4. Gastrin Vaccines

Pancreatic cancer cells have been shown to co-express gastrin and its receptor, which have been postulated to be involved in autocrine/paracrine signaling contributing to pancreatic cancer cell growth [197]. There have been multiple studies that have assessed the vaccine G17DT in PDAC, targeting the amino-terminal sequence of gastrin-17 (G-17), the bioactive form of gastrin. The vaccine also incorporates a spacer peptide linked to diphtheria toxoid to enhance immune responses. In preclinical studies, the vaccine was found to induce neutralizing antibodies against both amidated G-17 and its precursor molecule, glycine-extended G-17, inhibiting the growth of tumor cells [198]. Neutralizing antibodies targeting gastrin might prevent receptor interaction and disrupt the gastrin autocrine growth pathway.

In a phase II study of G17DT in patients with advanced pancreatic cancer, 67% of subjects produced an antibody response, including 82% of subjects at the higher dose level. Antibody responders had a significantly longer median OS compared to non-responders (217 days vs. 121 days; *p* = 0.023). Although three patients had serious adverse reactions, including recurrent abscess, fever, and pain requiring hospitalization, these three patients also exhibited above-average antibody responses and had a longer median OS compared to the entire group [17].

After a subsequent phase II trial of G17DT, using an accelerated dosing schedule was found to be immunogenic and well tolerated [18], a randomized, double-blind, placebo-controlled trial of G17DT was undertaken in PDAC patients who were unsuitable or unwilling to receive chemotherapy. In the intention-to-treat (ITT) population, there was no statistical difference in mortality between G17DT and placebo. However, patients who developed an antibody response to G17DT had a longer median OS than non-responders and patients who received a placebo (176 days vs. 63 days vs. 83 days, respectively; 95% CI: 142–227 vs. 28–150 vs. 71–120; log-rank test, *p* = 0.003) [16].

#### 4.1.5. Survivin Vaccines

Survivin, a member of the inhibitor of apoptosis family, plays a critical role in preventing apoptosis [199]. Its expression has been detected in up to 76% of PDAC and may be upregulated early in tumorigenesis [200]. SVN-2B, a peptide derived from the Survivin 2B splice variant, has been incorporated into vaccine trials targeting advanced PDAC.

A vaccination against survivin-2B peptide, along with Freund’s adjuvant (IFA) and IFNα, was tested in advanced PDAC patients in a small phase I trial. Type 1 IFNs (α and β) are known to have a variety of immunological effects, including increasing cross-presentation in dendritic cells [201]. The treatment was well-tolerated, with four out of six patients having a response to the vaccine, as well as a favorable clinical response of stable disease (SD) via computed tomography (CT) evaluation [19].

In a subsequent phase II trial, 83 patients with advanced PDAC were randomly assigned to one of three treatment arms: (I) SVN-2B and IFNβ, (II) SVN-2B alone, or (III) placebo. IFNβ was chosen instead of IFNα due to its ability to enhance CTL responses at lower doses while being better tolerated. There was no difference in PFS across the three groups, nor was there a difference in OS in the exploratory analysis. A small number of patients went on to receive four additional doses of SVN-2B and IFNβ (“Step 2”) after their initial treatment (“Step 1”). Among them, those who received SVN-2B and IFNβ in Step 1 had a better OS compared to those initially assigned to the placebo group. This may suggest that a longer vaccination protocol could have been more beneficial, although further studies would be needed to test this hypothesis [20].

### 4.2. Dendritic Cell Vaccines

Dendritic cells (DCs) are specialized antigen-presenting cells (APCs) that play a crucial role in T-cell priming and activation after their maturation and migration into lymphoid organs. They are potent APCs that can prime naïve T cells and induce the formation of effector and memory T cells targeting specific antigens. Taking advantage of this process, DC vaccines are created by isolating a patient’s dendritic cells from the peripheral blood. The DCs are then exposed to tumor antigens ex vivo, typically tumor peptides or tumor cell lysates, in a process known as “pulsing”. While tumor lysates may provide exposure to a broader range of tumor-specific and tumor-associated antigens with a more efficient induction of cancer-specific CTLs, this requires the patient’s tumor samples and presents greater challenges for quality control. The DCs process and present these antigens on their surface via major histocompatibility complex (MHC) molecules. Once infused into the patient, these activated DCs prime and activate CD4^+^ (helper) and/or CD8^+^ (cytotoxic) T cells, enabling them to target tumor cells possessing the same antigens.

In an early phase I/II trial of five patients with advanced PDAC harboring RAS mutations, peripheral blood mononuclear cells (PBMCs) were collected through leukapheresis and then pulsed with a synthetic version of the mutated peptide. The peptide-loaded cells were washed and reinfused multiple times, with the intention of trafficking to lymphoid organs to prime and activate T cells. The study found the vaccine to be safe, and two of the five patients developed a transient T-cell response to the RAS-mutated peptide [21]. In another phase I trial, DCs were isolated from the peripheral blood of patients with advanced pancreatic cancer and pulsed with a MUC-1 peptide. The vaccine was administered intradermally, as this administration route is more effective for DC trafficking to the draining lymph nodes, where antigen presentation can occur [202]. TNFα was used to promote the maturation of DCs, a requirement for them to be able to activate naive T cells. The vaccine was well tolerated, with two out of seven patients producing a T-cell response, quantified by IFNγ and granzyme B production by PBMCs. No significant clinical responses were observed [22].

A WT1 peptide-pulsed DC vaccine was tested in a pilot phase I trial in eight patients with resected pancreatic cancer. After surgery, patients received the WT1-DC vaccine combined with S-1 chemotherapy (an oral fluoropyrimidine) or S1 and gemcitabine. The adjuvant, OK-432, was also used. There were no serious side effects, and there was a high rate of immunological induction, with WT1-specific CTLs observed in seven out of eight patients [23].

In a study of mostly pretreated patients with unresectable advanced pancreatic cancer, participants received a WT-1 peptide-pulsed DC vaccine with or without other peptides, including MUC-1, CEA, or CA-125. Most patients also received lymphokine-activated killer (LAK) cell therapy, which has been shown to induce the phenotypic maturation of DCs in co-culture [203]. The treatment was combined with chemotherapy, including gemcitabine and/or S-1. The therapy was well tolerated, and some patients showed increased numbers of antigen-specific CTLs and decreased suppressive Tregs. A reduction in Tregs was associated with longer OS, a finding which may also be associated with the effects of gemcitabine, as has been reported previously [204]. The median OS was 360 days, with a notable benefit compared to historical controls treated with gemcitabine or 5-FU alone. Two patients achieved a complete response (CR), five had partial remission (PR), and ten had stable disease (SD) [24].

A recent study highlighted the potential of a WT1-DC vaccine combined with a single cycle of gemcitabine and nab-paclitaxel to convert unresectable PDAC into surgically resectable disease. The WT1 peptide cocktail included a multifunctional helper peptide targeting MHC class II and WT1-restricted CD4^+^ helper T-cell and CD8^+^ cytotoxic T-cell epitopes. In a phase I trial of 10 patients with unresectable PDAC (6 with locally advanced, 3 with metastatic, and 1 with recurrent disease), subjects received one cycle of gemcitabine and nab-paclitaxel, followed by 15 doses of the WT1-DC vaccine. The regimen was well tolerated, and notably, 8 out of 10 patients became eligible for resection, with 7 achieving R0 resection. Patients with a sustained WT1-specific immune response, indicated by long-term WT1-DTH positivity (*n* = 3), had significantly better outcomes, with a notable OS of at least 4.5 years. These patients also had a significantly higher percentage of WT1-specific CD4^+^ and CD8^+^ T cells producing IFNγ or TNFα, along with lower percentages of immunosuppressive populations [25].

Another recent study evaluated a DC vaccine loaded with allogeneic mesothelioma cell lysate for use in the adjuvant setting. In a phase I/II trial, 38 patients with resected pancreatic cancer, following standard-of-care treatment, received multiple doses of the vaccine. The PREOPANC trial, which included patients who received neoadjuvant chemoradiation, resection, and adjuvant chemotherapy with resectable pancreatic cancer, reported a 2-year RFS rate of 40%. Based on these results, the researchers hypothesize that adding adjuvant DC therapy could improve RFS from 40% to 60%. The estimated 2-year RFS observed in the study was 64%, suggesting a potential benefit of the vaccine. The expansion cohort also demonstrated the activation of CD4^+^ helper T cells in the peripheral blood [26].

Other studies have examined various DC vaccine approaches, including OK432-pulsed DCs administered intratumorally with LAK cells stimulated with an anti-CD3 monoclonal antibody (CD3-LAKs) and gemcitabine [27], α-Gal epitope-expressing tumor cell-pulsed DCs [28], DCs pulsed with MHC-I/II-restricted WT1 epitopes [29], and an hTERT, CEA, and survivin peptide-pulsed DC vaccine with a toll-like receptor (TLR)-3 adjuvant [30]. Similarly to the other DC vaccine studies in pancreas malignancy, immunological induction was observed in some patients in these studies, but this did not result in a significant antitumor response. It has been suggested that improving the standardization of DC vaccine production across centers—such as preparation or maturation strategies—could advance research. Additionally, combining DC vaccines with other immunomodulatory agents, targeting multiple DC subsets, using nanoparticle delivery systems, and incorporating DC-derived exosomes are potential strategies to further improve clinical outcomes [30].

### 4.3. Whole-Tumor-Cell Vaccines

Whole-tumor-cell vaccines utilize entire tumor cells—either autologous (derived from the patient) or allogeneic (from a cell line)—in their formulation. One potential benefit is the exposure to a broad range of TAAs and TSAs to help induce antitumor immunity. While autologous vaccines potentially offer the full array of TAA/TSAs specific to a patient’s tumor, compared to an allogeneic approach, they may be limited by challenges such as the availability of adequate tissue, as well as higher costs and the complexity of individualized production. Whole-tumor-cell vaccines are irradiated prior to administration to ensure safety.

#### 4.3.1. GVAX-Based Vaccines

GVAX was the first whole-tumor-cell vaccine to be used in PDAC. It was constructed using two separate irradiated human pancreatic cell lines, PANC 10.05 and PANC 6.03. To enhance its immunogenicity, both cell lines were genetically modified with a plasmid vector to secrete GM-CSF [31]. In an initial phase I study, PDAC patients received GVAX after surgical resection, followed by adjuvant therapy, and additional doses of GVAX if they remained in remission. The vaccine was found to be safe, with most patients experiencing only mild local toxicity at the injection site. In three patients, the vaccine successfully triggered antitumor immunity, measured by postvaccination DTH responses against tumor cells. These responses were associated with extended DFS [31].

Building upon these promising early results, several more studies investigated GVAX. In a single-arm study, patients received GVAX after surgery, followed by 5-FU-based chemoradiation, with additional GVAX boosters administered if they remained in remission. The vaccine was well tolerated, and OS compared favorably to the standard of care at the time. Notably, consistent with earlier findings, the induction of CD8^+^ T-cell responses targeting multiple mesothelin epitopes correlated with improved DFS [205].

Another study evaluated GVAX with low-dose cyclophosphamide, an alkylating chemotherapy agent, in the neoadjuvant setting to primarily evaluate for changes to the tumor microenvironment (TME) of PDAC. The incorporation of low-dose cyclophosphamide is based on preclinical studies demonstrating its ability to deplete suppressive Tregs [206]. While the study was not powered to compare the efficacy of treatment arms, patients did not appear to derive additional benefit from low-dose cyclophosphamide compared to GVAX alone. The study did provide important insights into the TME, showing that GVAX promotes the formation of tertiary lymphoid structures (TLSs), within the tumor. These ectopic lymph node-like structures were found in higher density in patients with prolonged survival [34]. While their significance here is not fully understood, tertiary lymphoid structures are strongly associated with better clinical outcomes following immune checkpoint blockade for various cancers [207].

In another study, GVAX was evaluated in combination with low-dose cyclophosphamide, SBRT, and modified FOLFIRINOX in the adjuvant setting. The multi-agent regimen was tolerable based on phase I data, although no significant conclusions regarding efficacy could be made given the early-stage design [33]. In the metastatic setting, GVAX has also been combined with the CTLA-4 blocking antibody ipilimumab [35] in a phase I study and with CRS-207 and cyclophosphamide in a phase II study. CRS-207 is a live, attenuated *Listeria monocytogenes* engineered to express mesothelin. Although well tolerated, the GVAX, cyclophosphamide, and CRS-207 investigational arm did not show a benefit in OS over standard chemotherapy [36].

#### 4.3.2. Algenpantucel-L (Hyperacute–Pancreatic Cancer Vaccine)

Another whole-tumor-cell vaccine to be used in PDAC is algenpantucel-L, also known as HyperAcute-Pancreas cancer vaccine. The vaccine consists of two pancreatic cancer cell lines (HAPa-1 and HAPa-2) engineered to express α(1,3)-galactosyl epitopes on tumor cells. The process is meant to trigger a hyperacute immune response, similarly to the hyperacute rejection seen in organ transplantation via anti-αGal antibodies.

In the phase II study, patients safely received algenpantucel-L with adjuvant chemotherapy and chemoradiation, comparing favorably to historical adjuvant data [37]. However, the phase III trial did not show an improvement in PFS or OS when the vaccine was added to standard-of-care chemotherapy or chemoradiation in the neoadjuvant setting for patients with borderline resectable or locally advanced PDAC [38].

### 4.4. Nucleic Acid-Based Vaccines and Personalized Neoantigen Vaccines

Personalized neoantigen vaccines are created by identifying unique neoantigens within an individual patient’s tumor. While offering the possibility of a vaccine that is more specific to a particular individual, there are several key considerations. The most important factor is whether identified mutations are processed, presented on MHC molecules, and elicit functional T-cell responses. Additionally, the ability to identify candidate neoantigens and manufacture and deliver the vaccine quickly and inexpensively has been another major factor. However, new sequencing and mRNA vaccine technologies are solving this.

An initial study assessing the neoantigen landscape in PDAC found that nearly all tumor samples have potentially targetable neoantigens, including ones expected to have efficient presentation by MHC class I molecules. However, while these patients’ tumor samples possessed tumor-infiltrating lymphocytes (TILs) within the TME, there was an overall immunosuppressive tumor microenvironment with a reduced expression of activated transcripts by T cells [123]. Another study found that while some PDAC patients have a higher neoepitope burden, this did not necessarily correlate with higher intratumoral cytolytic T-cell activity. Notably, there were patients with low neoepitope burden yet high intratumoral cytolytic T-cell activity, while others with a high burden showed minimal immune activation [208].

A more recent study found that while PDAC tumors with the highest neoantigen number did not necessarily correlate with an outcome of longer survival, the study was able to demonstrate a method for identifying high-quality neoantigens capable of inducing intratumoral T-cell reactivity. The model prioritized neoantigens with both differential presentation patterns and homology similar to pathogen-derived peptides. Additionally, the study found that neoantigens from certain “immunogenic hotspots”, such as the MUC16 locus, were more common in long-term survivors with PDAC.

To apply this bioinformatics model of neoantigen selection clinically, the investigators conducted a notable phase I trial in patients with resected PDAC. Somatic mutations in patients’ tumors were identified by next-generation sequencing, and then, a list of immunogenic neoantigens was predicted. Each patient’s vaccine was constructed by identifying up to 20 neoantigen epitopes, delivered as two messenger RNAs encoding up to 10 neoepitopes each. The vaccine, known as autogene cevumeran, was delivered as a lipoplex nanoparticle to facilitate IV delivery, protect the RNA from degradation, and more effectively target APCs in the spleen and other lymphoid organs. Additionally, the RNA molecule works as an adjuvant stimulus, activating toll-like receptors (TLRs) 7 and 8, found on APCs.

Participants with surgically resected PDAC sequentially received atezolizumab (an anti-PDL-1 antibody), autogene cevumeran, and mFOLFIRINOX (a chemotherapy regimen consisting of folinic acid, fluorouracil, irinotecan, and oxaliplatin). Administering a single dose of atezolizumab was felt to be adequate and potentially beneficial, as receptor occupancy from a single dose of PD-L1 inhibition can persist for several months. Of the 16 patients treated, 8 exhibited the production of high-magnitude neoantigen-specific T-cell responses. After a median follow-up of 18 months, the same 8/16 patients with significant responses to the vaccine also had a longer median RFS compared to those without significant responses (not reached vs. 13.4 months, *p* = 0.003) [39].

In an extended median follow-up of 3.2 years, responders maintained durable T-cell responses, correlating with clinical benefit. In responders, these T-cell clones had an average estimated lifespan of 7.7 years, with some expected to persist for decades. In addition, most of these clones transitioned into memory T cells without showing features of exhaustion, resembling a tissue-resident memory-like T-cell state and potentially contributing to long-term immune surveillance [209].

### 4.5. Viral or Bacterial Vector-Based Vaccines

Several studies have evaluated viral or bacterial vector-based vaccines in PDAC. Engineered viruses and bacteria can encode tumor antigens and can be modified with genetic elements to enhance their therapeutic effects. In addition to stimulating immune activation, viruses may also exert antitumor activity by direct oncolysis, infecting, replicating within, and destroying tumor cells and associated stromal cells. This can potentially release tumor antigens and new viral particles, triggering further immune activation.

In a randomized, open-label phase II trial, patients with advanced PDAC were assigned to receive gemcitabine with or without IMM-101. IMM-101 is a suspension containing heat-killed whole-cell *Mycobacterium obuense* meant to enhance immune responses. The vaccine was safe and well tolerated, with a statistically significant benefit in OS in the predefined metastatic subgroup [40].

VXM01 is a novel, first-in-class vaccine that is administered orally. It uses a live, attenuated *Salmonella* bacteria to deliver a eukaryotic expression plasmid encoding the VEGFR-2 protein. VEGFR-2 is primarily expressed on tumor vasculature, as well as on certain tumor cells, where it responds to growth and survival signals from VEGF-A, promoting angiogenesis. In a randomized, dose-escalation phase I trial in patients with locally advanced or metastatic pancreatic cancer, *Salmonella* excretion and *Salmonella*-specific humoral immune responses occurred in the two highest dose groups. Vaccinated patients showed increased VEGFR-2-specific effector T-cell responses, with Treg responses remaining unchanged. A significant reduction in tumor perfusion was observed 38 days after vaccination, indicative of an anti-angiogenic immune response. Additionally, patients with pre-existing VEGFR-2-specific T cells showed a strong correlation with greater reductions in tumor perfusion, indicating that the reactivation of pre-existing memory T cells may, at least in part, contribute to vaccine efficacy [41].

In a phase I trial, patients with advanced PDAC were given heterologous prime-boost vaccination with poxviruses expressing both CEA and MUC-1. The vaccines also expressed three costimulatory molecules—B7.1, ICAM-1, and LFA-3—known as TRICOM, which have been shown to enhance T-cell activation [210]. GM-CSF was used as an adjuvant after each vaccination. Antigen-specific T-cell responses were observed in 62.5% of evaluable patients, and antibody responses occurred in all patients. Those with immune responses to CEA and/or MUC-1 had a significant increase in OS [42]. However, the exploration of this vaccine platform slowed when a phase III trial of >1000 patients with advanced prostate cancer revealed no benefit of a PSA-directed TRICOM vaccine with or without GM-CSF compared to the placebo [211].

A proprietary formulation of an oncolytic reovirus was evaluated in a randomized phase II trial in combination with carboplatin and paclitaxel for advanced PDAC. The virus has direct cytotoxicity of tumor cells with RAS pathway activation [212], with the potential to also enhance antitumor immune responses [213], although conflicting evidence also suggests that it may contribute to immunosuppression [214]. The study did not demonstrate an improvement in PFS with the vaccine compared to chemotherapy alone. Additionally, patients receiving the vaccine had an increased expression of the checkpoint molecule CTLA-4 on both CD4^+^ and CD8^+^ T cells. Vaccinated patients also had increased levels of immunosuppressive markers (e.g., IL-6, VEGF, Tregs), which potentially may have been either promoted by the vaccine or upregulated as a secondary response to T-cell activation after antigen exposure. Regardless of the mechanism, the vaccine was unable to overcome suppressive TME changes in PDAC to provide clinical benefit [43].

## 5. Key Areas of PDAC Cancer Vaccine Research

There are many ongoing clinical trials of cancer vaccines in PDAC across various contexts, including resectable or borderline resectable, locally advanced, unresectable, metastatic, and preventative settings (Table 2). A major focus of study is the post-surgical setting, where disease burden is low and micrometastatic disease can be addressed [215,216,217,218,219,220,221,222,223,224,225,226,227]. The timing of vaccination relative to adjuvant chemotherapy varies among these trials, with some administering vaccines after adjuvant chemotherapy [215,223,224,226,227] and others choosing to immunize beforehand [219,220]. The incorporation of immune checkpoint blockade to augment the immune response has also been widely adopted and is now standard in most ongoing studies [217,218,219,220,221,222,224,225,226,228,229,230,231,232,233,234,235]. Other immune modulatory strategies involve the use of CD40 or CD137 agonists [228,229] or the CD38 monoclonal antibody, daratumumab [234]. Additionally, vaccination in the window prior to surgery (neoadjuvant setting) is being further explored [228,236].

There is an emphasis on developing personalized neoantigen vaccines, particularly in the resectable and borderline resectable setting. These utilize a variety of platforms, including peptide-based [215,216,229,236], mRNA-based [217,218,219,220,221,230], dendritic cell-based [222], and yeast-based approaches [223]. The mRNA platform is increasingly being utilized across studies, having garnered attention during the COVID-19 pandemic due to its production speed and safety [237], as well as its high feasibility to create personalized mRNA vaccines compared to other platforms. Several shared neoantigen vaccines are being tested in trials, with mKRAS as the most common target [225,231,233,234]. Other targets include WT1 [238], CEA and MUC1 [239], and p53 [235]. There is also interest in developing vaccines for primary prevention for those at high risk of developing PDAC [233].

## 6. Other Immunotherapy Approaches

Aside from vaccines, several immunotherapeutic approaches are currently under study or undergoing further exploration. The traditional immune checkpoint blockade (ICB) strategies targeting the PD1/PDL1 axis or CTLA-4 have not yielded impressive results in pancreatic cancer therapy compared to responses observed in other solid tumors [240]. The results from the KEYNOTE-158 trial led to the tumor-agnostic approval of single-agent pembrolizumab in tumors with either a high TMB or mismatch repair deficiency; however, this patient population accounts for only about 1% of PDAC cases, limiting the application of pembrolizumab in PDAC [159,241]. Moreover, other tumor microenvironment factors may also limit PDAC immunity through the suppression of Th1 CD4^+^ T-cell responses and/or CD8^+^ cytotoxic T-cell responses, the recalibration of the milieu to favor tumor growth, and uniquely severe desmoplasia, which makes the TME fibrotic and creates physical barriers, making the delivery of cytotoxic chemotherapy or immune infiltration into the PDAC TME difficult. While these challenges remain largely unsolved, other strategies that have been explored include targeting novel immune checkpoints in the TME, adoptive cell therapy, oncolytic virus therapy (OVT), the targeting of myeloid cell populations, and the modulation of the microbiome, among others.

### 6.1. Oncolytic Virus Therapy

Oncolytic virus therapy is a promising area of research in which naturally equipped or genetically engineered viruses are employed to selectively replicate in tumor cells, inducing cell lysis and the release of tumor antigens. Pathogen-associated molecular patterns (PAMPs) and damage-associated molecular patterns (DAMPs) induce innate cell activation and the presentation of released tumor antigens to T cells to induce de novo T-cell responses and/or boost pre-existing responses [242]. Pathogen-associated molecular patterns (PAMPs) and damage-associated molecular patterns (DAMPs) induce innate cell activation and the presentation of released tumor antigens to T cells to induce de novo T-cell responses and/or boost pre-existing responses [242]. Additionally, these viruses can also be engineered to produce additional proinflammatory cytokines, which are released into the TME to further augment the ability of OVTs to induce tumor-specific immune responses. The first, and only, FDA-approved OVT is intralesional Talimogene laherparepvec (“T-VEC”; brand name IMLYGIC) for recurrent melanoma lesions after prior surgery based on results of the OPTiM study [243]. Several other DNA and RNA viruses have been explored for this purpose in preclinical and phase I/II clinical studies, with some notable examples below.

#### 6.1.1. VCN-01

VCN-01 is an oncolytic adenovirus that replicates specifically in cells that have a defective RB-1 signaling pathway. This virus also expresses hyaluronidase to degrade hyaluronic acid in the extracellular matrix to counteract the robust desmoplastic TME of PDAC and improve the penetration of chemotherapeutic agents or immune cells in PDAC [244]. As such, a phase 1 trial has been conducted to assess safety and to determine the recommended dose of VCN-01 for phase 2 studies (RP2D) given via intravenous administration along with nab-paclitaxel and gemcitabine. The results generally concluded that VCN-01, in conjunction with chemotherapy, was safe and paved the way for further studies. An ORR of about 50% was reported as well [245].

#### 6.1.2. ONYX-015

ONYX-015 is an oncolytic adenovirus that selectively replicates in cells that have p53 deficiency. Following a successful phase 1 trial [246], a follow-up phase 1/2 trial examined the endoscopic ultrasound (EUS)-guided intratumoral administration of ONYX-015. ONXY15 was administered eight times, with the last four cycles supplemented with gemcitabine. The study revealed the safety and feasibility of direct adenoviral delivery to PDAC tumors via EUS; however, intratumoral ONYX-015 in PDAC displayed limited efficacy [247].

#### 6.1.3. LOAd703

LOAd703 is an oncolytic adenovirus that encodes trimerized, membrane-bound (TMZ)-CD40L and 4-1BBL, which are intended to provide robust costimulatory signals to cytotoxic T cells and induce tumor regression. A non-randomized phase 1/2 trial (LOKON001) is currently underway, and results from arm 1 of the trial were used to evaluate the safety of intratumoral LOAd703 injections in combination with standard chemotherapy. The findings demonstrated the safety and tolerability of the injection, with the highest given dose being tolerated safely. Arm 2 of this trial evaluated further efficacy and included the addition of atezolizumab to the above combination of treatments [248].

#### 6.1.4. Palareorep (Reolysin)

Palareorep (Reolysin) is a proprietary isolate of unmodified human reovirus that has been evaluated in a phase II study in combination with conventional chemotherapy. While it was generally safe, it did not improve PFS when compared to standard chemotherapy alone [43].

#### 6.1.5. Others

Other notable examples being explored include H101, an oncolytic adenovirus targeting malignant ascites; HF10, an attenuated, replication-competent mutant strain of Herpes Simplex Virus (HSV) type 1 in locally advanced pancreatic cancer; and Ad5-yCD/*mut*TK_SR39_*rep*-hIL-12, an oncolytic adenovirus expressing cytosine deaminase and thymidine kinase to convert non-toxic pro-drugs to DNA synthesis inhibitors and human IL-12 to promote responses by NK cells, Th1 CD4^+^ T cells, and cytotoxic CD8^+^ T cells [249,250].

### 6.2. Adoptive Cell Transfer

Adoptive cell therapies (ACTs) involve the transfer of cancer-targeting immune cells, such as T cells, to the patient to mediate antitumor immunity. These are generally cytotoxic cells targeting tumors, resulting in the direct targeting and lysis of cancer cells and tumor ablation. Adoptive cell therapies, particularly CAR-T cells, have made great strides in the treatment of hematologic malignancies, with multiple FDA-approved CAR-T products being used in relapsed/refractory disease and now under evaluation for frontline therapy. The exploration of ACT began in solid tumors, far in advance of CAR-T cell success in hematologic malignancies, but successful application was not seen until 2024 with FDA approvals for liflileucel (AMTAGVI), a tumor-infiltrating lymphocyte (TIL) therapy for unresectable or metastatic melanoma, and afamitresgene autoleucel (Tecelra), a MAGE-A4 TCR-engineered T cell therapy for unresectable or metastatic synovial sarcoma. In that context, several TIL, CAR-T, and other ACT approaches are being explored for PDAC.

#### 6.2.1. Tumor Infiltrating Lymphocytes (TILs)

The quantity and organization of tumor-infiltrating lymphocytes (TILs) are strong prognostic indicators in various cancers, as well as for immune checkpoint blockade efficacy. Beyond being a prognostic tool, it is also possible to utilize these TILs therapeutically. This type of therapy involves the ex vivo culturing and expansion of antigen-specific T lymphocytes from the patient’s own resected tumor (TILs). This is then followed by infusion back to the patient after nonmyeloablative chemotherapy and (typically) IL-2 administration to improve TIL expansion and persistence in the patient [251,252]. Studies from pancreatic tumor excision specimens provided data that TILs are present in the TME and are capable of recognizing tumor antigens, suggesting that they can be potentially expanded to yield therapeutic TILs for therapy [251]. Studies from pancreatic tumor excision specimens provided data showing that TILs are present in the TME and are capable of recognizing tumor antigens, suggesting that they can potentially be expanded to yield therapeutic TILs for therapy [253]. The results from the ongoing NCT01174121 trial suggest the robust activity of neoantigen-specific TILs in the >50 treated patients with colorectal cancers [254]. Moreover, a patient with PDAC metastatic to the liver, lymph nodes, and peritoneum experienced a partial response. Another PDAC patient experienced the complete regression of dozens of liver metastases, some of which recurred from potential antigen loss. These results are highly promising, but the limited number of PDAC patients treated with TILs in that trial potentially highlights some of the challenges with TIL-based therapies in this patient population.

#### 6.2.2. CAR-T Cell Therapy

CAR-T therapy involves the collection of autologous T cells from the patient, genetically modifying them to express a chimeric antigen receptor (CAR) directed against an antigen expressed on the surface of cancer cells, expanding them ex vivo, and administering them back to the patient [255]. Antigenic targets evaluated for CAR-T for pancreatic cancer in preclinical or clinical models notably include prostate stem cell antigen, mesothelin (MSLN), CEACAM7, CEA, HER2, Mucin-1, FAP, and others [256,257,258,259,260,261]. None of these have yet resulted in sufficient efficacy with acceptable toxicity, and CAR-T therapy, in general and in the context of pancreatic cancer, faces concerns of antigen escape (particularly in the context of single-antigen-targeted CAR-T cells), concerns regarding inadequate tumor infiltration, and concerns for on-target/off-tumor toxicity [262]. However, enthusiasm for PDAC CAR-T therapy remains high as CAR-T technologies to overcome these barriers continue to advance, and additional antigen targets are identified and explored [262]. However, enthusiasm for PDAC CAR-T therapy remains high as CAR-T technologies to overcome these barriers continue to advance, and additional antigen targets are identified and explored.

#### 6.2.3. CAR-NK Cell Therapy

CAR-NK cells are another strategy being evaluated, offering some benefits compared to CAR-T cells, including reduced cytokine release syndrome (CRS). Moreover, allogeneic NK cells have limited ability to cause graft-versus-host disease (GVHD) compared to T cells, creating the potential to use mass-manufactured, donor-derived CAR-NK cells rather than autologous CAR-T cells. However, difficulties with donor selection, antigen heterogeneity, difficulties in designing effective CAR-NK cell products, and, most importantly, the limited in vivo expansion and persistence of CAR-NK cells limit this approach [263]. Targets that have been evaluated include ROBO1, MUC-1, PSCA, and MSLN [264,265]. CAR-NK products derived from induced pluripotent stem cells (iPSCs) are also currently undergoing phase 1 evaluation (NCT03841110).

### 6.3. Therapies Targeting Immune Inhibitory Checkpoints

#### 6.3.1. PD1/PDL1, CTLA-4, and LAG-3

PD1/PDL1, CTLA-4, and LAG-3 are well-established inhibitory signaling axes in T cells in several cancers in animal models and patients, resulting in several FDA-approved therapies targeting these axes (immune checkpoint blockade; ICB). However, because targeting none of these alone has produced impressive results in PDAC, combinations with other therapies or immune checkpoints are under exploration. Notably, prior chemotherapy shows that chemotherapy-induced senescent cells can upregulate PDL2 expression. PDL2 is a ligand for not only PD1 but also RGMb, and PDL2-RGMb interactions may underlie the failure of PD1 blockade [266]. Moreover, the blockade of PDL2 in senescent cancer cells in combination with chemotherapy induces tumor regression in mouse models [267]. Other PD1/PDL1 combination targets that have been explored include CCL5, TNFR2, and others [268,269]. ADH-503 acts on CD11b and causes an agonistic effect on innate immune responses, which in turn make tumors more responsive to therapies such as PDL1 blockade and CD137 agonist therapies [270]. A bispecific antibody targeting PD1 and IL2Rβγ has been studied in combination with radiotherapy and can lead to reduced tumor growth and metastasis, enhancing NK cell immune response and the induction of durable responses [271]. The dual blockade of CTLA-4 and IL-6 has also been explored as a possible mechanism for improving immune responses [272]. Cells in pancreatic tumors have been shown to use LAG-3-directed signaling to create an immunosuppressive environment. As such, the antagonism of LAG-3 along with CD137 agonists has been explored together, and along with a CXCR1/2 inhibitor in myeloid cells, it can lead to durable responses in mouse models [273].

#### 6.3.2. Emerging Inhibitory Checkpoints

The increased expression of TIGIT, as well as PD1, has been observed on the surface of CD8^+^ tissue resident memory T cells (TRMs) when pancreatic tumor excision samples were analyzed. Blocking these two receptors/pathways simultaneously can lead to a synergistic effect in reversing immunosuppression in the TME in mice [274]. VISTA is a member of the B7 family of costimulatory/coinhibitory proteins that inhibits T cells and myeloid cells. Given this function, monoclonal antibodies against VISTA could potentially act as another functional approach [275]. CD39 and CD73 are both ectonucleotidases that cooperatively deplete ATP and increase adenosine in the TME, which in turn exerts immunosuppressive actions and promotes tumor growth, spread, and resistance to chemotherapy [276]. Several experimental antibodies, such as oleclumab and mupadolimab, have undergone phase 1 safety trials and are currently under phase 1b/2 or phase 2 trials for evaluation in pancreatic cancer [277,278]. OX40 (CD134/TNFRSF4) is a T-cell costimulatory receptor that belongs to the tumor necrosis factor receptor superfamily. Generally present on the surface of T cells, OX40 binds to its ligand OX40L on the surface of antigen-presenting cells to stimulate T-cell proliferation and survival [279]. OX40 agonists have been explored in mouse orthotopic pancreatic cancer models, along with the administration of anti-PD1 therapy, and they have led to a reduction in regulatory and exhausted T cells and the stimulation of memory CD4^+^ and CD8^+^ T cells and B cells, resulting in tumor eradication [280]. To date, however, evaluations in clinical trials have not yet yielded clear clinical responses.

### 6.4. Myeloid Cell-Directed Therapies

#### 6.4.1. CD40 Agonists

CD40 is a co-stimulatory receptor that is present on the surface of B cells, dendritic cells, and macrophages. It promotes the M1 phenotype in macrophages, which enhances their phagocytic and antigen-presenting capacities and increases proinflammatory cytokine secretion, resulting in the increased activation of antitumor T-cell responses [281]. More recently, a promising phase 1b/2 study (OPTIMIZE-1) was reported, with data from 70 patients across multiple European sites evaluating the CD40 agonist mitazolimab in conjunction with mFOLFIRINOX. RP2D was established, and patients had a median follow-up of 12.7 months. The trial achieved its primary endpoint, an overall response rate (ORR) of greater than 30%. Among 57 evaluable patients, objective responses were noted in 23 individuals, and 22 other patients had stable disease [282]. Among 57 evaluable patients, objective responses were noted in 23 individuals, along with 22 other patients having had stable disease [282].

Dendritic cells have been characterized as scarce and generally dysregulated in the milieu of PDAC, which leads to ineffective antigen presentation and T cell responses. Thus, targeting or restoring the function of dendritic cells can be a potential target for enhancing antitumor immunity [283]. In this regard, CD40 agonistic agents are being evaluated in trials (NCT03329950) in combination with FLT3L (CDX-301). Moreover, several other approaches to activate dendritic cells were discussed above in the context of vaccines.

#### 6.4.2. CD47 Blockade

CD47 on cancer cells functions infamously as a “don’t eat me” signal to SIRPα on the surface of macrophages, blocking their phagocytic function, and thus offers a potential therapeutic target, which in animal models has been shown to inhibit tumor growth [284]. Currently, PT-886, a bispecific Ab targeting CD47 and the tumor antigen CLDN18.2 in combination with chemotherapy and/or pembrolizumab, is under evaluation in a phase 1/2 clinical trial, which is actively recruiting participants with advanced gastric, GE junction, or pancreatic adenocarcinoma (NCT05482893).

#### 6.4.3. Other Emerging Modalities

CXCR2-related signaling is generally responsible for recruiting myeloid-derived suppressor cells (MDSCs) to the TME and, thus, fostering an immunosuppressive environment. Currently, a dual CXCR1/2 inhibitor, SX-682, is being evaluated in combination with LAG-3 blockade and CD137 agonists or in combination with PD1 blockade in the context of surgically resectable cancer [273]. The STING pathway is important for innate immune responses to pathogens, with cytosolic DNA from pathogens triggering the activation of STING receptors on the endoplasmic reticulum through cGAMP formation. The activation of STING receptors then leads to downstream signaling and the production of interferons and other proinflammatory cytokines to drive effective immune responses [285]. MARCO is a scavenger receptor that has been noted to be expressed on a subset of tumor-associated macrophages (TAMs), which limit antitumor immunity and confer poor prognosis. Antibodies directed against MARCO have been shown to either decrease the suppressive activity of TAMs or convert them into a more immunostimulatory phenotype [286].

## 7. Conclusions and Future Directions

PDAC remains a highly challenging malignancy to treat despite improved treatment paradigms and advances in multimodality therapy. In early-stage disease, high recurrence rates after surgery and adjuvant therapy underscore the need for novel therapeutic approaches [3,4,5]. Cancer vaccines offer one potential strategy that is particularly suited for the resectable or borderline resectable setting, where tumor burden is low and immunosuppression is less pronounced. While many vaccines have demonstrated safety and immunogenicity in early-phase trials, their clinical efficacy has been limited, highlighting persistent challenges such as tumor immune evasion, intratumoral heterogeneity, and poor immune cell infiltration.

Peptide vaccines have demonstrated safety and modest immune activation but have largely failed to show meaningful improvements in RFS, DFS, or OS [9,10,14,16]. For instance, the telomerase-targeted vaccine, GV1001, showed promise in an early-phase trial [15], yet it failed to show an OS benefit in the larger TeloVac study in advanced PDAC patients [14]. Similarly, gastrin-targeted vaccination with G17DT elicited antibody responses and improved outcomes in immunologic responders [17], but the phase III study did not show a clear benefit in advanced PDAC patients [16]. Strategies targeting various neoantigens, such as mKRAS [8,9,10,11,21], TERT [14,15,166], MUC1 [12,13], gastrin [16,17,18], and survivin [19,20], have overall shown immunogenicity in subsets of patients.

Dendritic cell vaccines and whole-tumor-cell vaccines have also shown immunogenicity in some patients but have lacked substantial clinical benefits. For instance, clinical trials of GVAX have shown immunological activity in generating CD8^+^ T-cell and DTH responses and promoting TLSs [31,32,34]; however, evaluating GVAX in combination with CRS-207 in a larger trial did not demonstrate significant clinical benefit [36]. Trials with GVAX remain ongoing, reflecting promising correlative data [228]. Similarly, algenpatucel-L failed to show a clinical benefit in a phase III trial despite early promise [38]. Personalized neoantigen vaccines are an emerging approach, with autogene cevumeran demonstrating feasibility and potential efficacy in PDAC [39], with many other personalized vaccines being evaluated in ongoing trials [215,216,217,218,219,220,221,222,223,229,230,236].

PDAC vaccines in combination with chemotherapy, radiotherapy, and/or immune checkpoint blockade are potential means of improving efficacy through potentially additive or synergistic effects. Optimizing the timing and sequence of vaccination relative to these cytotoxins (chemo/radiotherapy) or immunomodulators, as well as selecting appropriate adjuvants and booster schedules, should be further studied. Emerging biomarkers, such as ctDNA, will help enhance patient selection and response monitoring.

Aside from vaccines, other immunological strategies present novel ways of harnessing the immune response. These include oncolytic virus therapies, adoptive cell transfer, macrophage-directed interventions, and other approaches. Moreover, highly encouraging clinical data [287] and ongoing phase III studies (NCT05933577) examining personalized mRNA vaccines with ICB for melanoma recurrence prevention provide further optimism for the application of vaccines in other cancers, including PDAC. Though most remain in early phase trials, continued refinement and advances in existing modalities—including vaccines—may ultimately improve the treatment landscape.

## Figures and Tables

**Figure 1 cancers-17-02356-f001:**
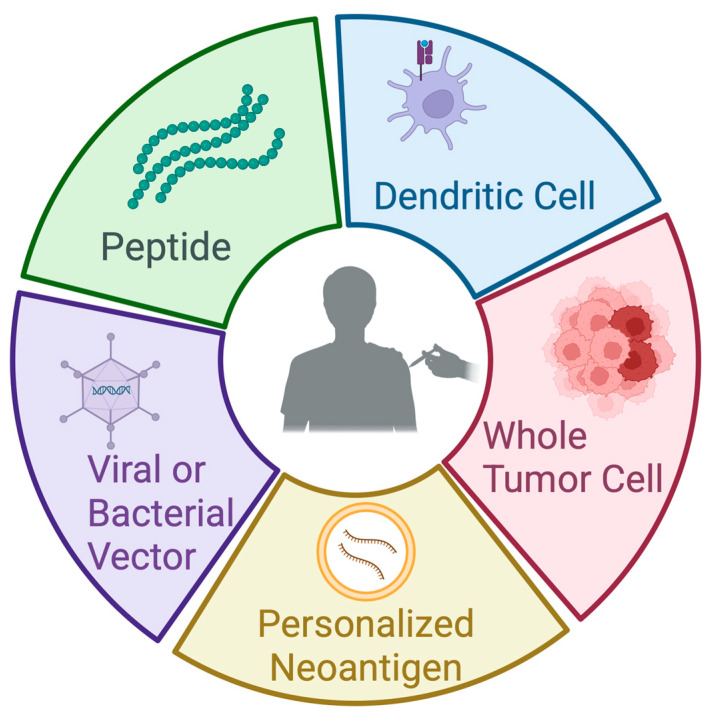
Types of vaccines in pancreatic cancer. This figure illustrates the primary platforms used in the development of pancreatic cancer vaccines. Peptide vaccines consist of short, synthetic fragments derived from tumor-associated antigens, designed to elicit a targeted immune response. Dendritic cell vaccines involve the ex vivo loading of patient-derived dendritic cells with tumor antigens, followed by reinfusion to stimulate T-cell activation. Whole-tumor-cell vaccines utilize irradiated or lysed autologous or allogeneic tumor cells to present a broad array of antigens to the immune system. Personalized neoantigen vaccines are peptide or mRNA vaccines based on sequencing data to target tumor-specific mutations unique to an individual’s cancer. Viral or bacterial vector-based vaccines deliver tumor antigens using engineered microbial platforms to enhance antigen presentation and immunogenicity. Each approach aims to promote a robust and specific antitumor immune response. Created in BioRender.

**Figure 2 cancers-17-02356-f002:**
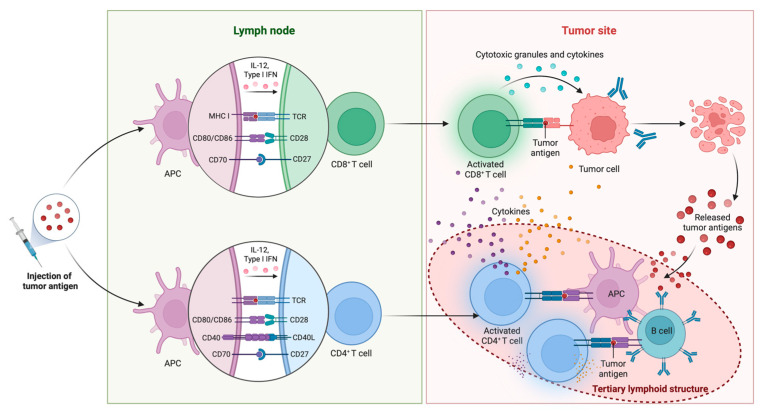
Mechanistic basis for vaccine-based immunotherapy. Following tumor antigen injection, APCs capture, process, and present the antigens on MHC molecules in lymph nodes, activating naïve CD8^+^ and CD4^+^ T cells. T cell activation is mediated through the TCR recognition of peptide-MHC complexes and costimulatory interactions (CD28-CD80/CD86, CD27-CD70, and CD40L-CD40), and it is supported by cytokines such as IL-12 and type I interferons (IFNs). Activated CD8^+^ T cells migrate to the tumor site, where they recognize antigen-expressing tumor cells and exert cytotoxic effects through the release of granules and proinflammatory cytokines, resulting in tumor cell lysis and the release of additional tumor antigens. This antigen release may support epitope spreading and the development of tertiary lymphoid structures (TLSs) within the tumor microenvironment that maintain tumor immunity through tumor eradication. Created in BioRender.

**Table 1 cancers-17-02356-t001:** Selected completed studies of cancer vaccines for pancreatic cancer.

	**Cancer Vaccine**	**Summary**	**Reference**
Peptide Vaccines	KRAS		
	Multi-KRAS peptide vaccine + GM-CSF	4/10 patients had extended DFS in adjuvant setting; in advanced disease, DTH activity and/or T-cell responses associated with improved OS	[8]
	TG01 vaccine (synthetic RAS peptides) + GM-CSF + adjuvant gemcitabine	High levels of immune activation; DFS and OS similar to adjuvant gemcitabine alone	[9]
	GI-4000 vaccine (mutant RAS proteins) + gemcitabine	No improvement in RFS compared to gemcitabine + placebo	[10]
	ELI-002 2P vaccine (amphiphilic KRAS peptides + Amph-CpG-7909 adjuvant)	84% of participants with KRAS-specific T-cell response and biomarker response; 24% achieved biomarker clearance (3 w/PDAC	[11]
	MUC-1		
	MUC1 peptide vaccine + BCG adjuvant	Well tolerated; minority of patients with immune responses	[12,13]
	Telomerase		
	GV1001 + GM-CSF + chemotherapy (phase III tested sequentially or concurrently with chemotherapy)	38% and 37% had immune responses in sequential and concurrent groups, respectively; no OS benefit	[14,15]
	Gastrin		
	G17DT	In phase III, patients with antibody response had significantly longer OS (176 vs. 63 vs. 83 days, *p* = 0.003); no OS benefit in ITT analysis	[16,17,18]
	Survivin		
	Survivin-2B80-88 peptide ± IFA/IFNα/IFNβ	Safe and immunogenic; no OS benefit, but extended dosing with IFNβ improved OS compared to initial placebo	[19,20]
Dendritic Cells	mKRAS peptide-pulsed DC vaccine	Safe; 2/5 patients developed a transient T-cell response	[21]
	MUC-1 peptide-pulsed DC vaccine	Well tolerated; 2 of 7 patients had increased IFNγ and granzyme B levels from PBMCs, but no significant clinical responses	[22]
	WT1 peptide-pulsed DC vaccine + OK-432 adjuvant + chemotherapy	No serious side effects; WT1-specific CTLs observed in 7/8 patients	[23]
	WT1 peptide-pulsed DC vaccine ± LAK cell therapy + chemotherapy	Well tolerated; increased antigen-specific CTLs and reduced Tregs associated with longer OS	[24]
	WT1-DC vaccine + chemotherapy	Well tolerated; 8/10 participants became eligible for resection; those with sustained WT1-specific immune response (*n* = 3) had significantly better outcomes, with OS ≥ 4.5 years	[25]
	DC vaccine loaded with allogeneic mesothelioma cell lysate	Estimated 2-year RFS of 64%; expansion cohort showed activation of CD4+ helper T cells in peripheral blood	[26]
	Various other DC vaccine strategies	Immunological responses, but no significant antitumor effect	[27,28,29,30]
Tumor Cells	GVAX		
	GVAX + adjuvant therapy	Safe; extended DFS in responders	[31,32,33]
	GVAX ± low-dose cyclophosphamide	Promoted tertiary lymphoid structures (TLSs); higher TLS density correlated with prolonged OS	[34]
	GVAX + ipilimumab	Increase in peak mesothelin-specific T cells among patients with OS > 4.3 months.	[35]
	GVAX + CRS-207 (Listeria monocytogenes expressing mesothelin) + cyclophosphamide	Well tolerated; no benefit in OS over standard chemotherapy	[36]
	Algenpantucel-L (Hyperacute–Pancreatic Cancer Vaccine)		
	Algenpantucel-L + chemotherapy/chemoradiation	Phase II with favorable comparison to historical data; phase III without improvement in PFS or OS	[37,38]
Persona Neoantigen	Autogene cevumeran + atezolizumab + mFOLFIRINOX	50% of participants with high magnitude neoantigen specific T-cell responses; significant responders with longer median RFS	[39]
Viral or Bacterial Vector	Gemcitabine ± IMM-101 (heat-killed *Mycobacterium obuense*)	Safe with statistically significant OS benefit in predefined metastatic subgroup	[40]
	VXM01 (oral vaccine with live, attenuated salmonella delivering VEGFR-2 plasmid)	Salmonella specific immune responses in higher dose groups; significant tumor perfusion reduction 38 days after vaccination	[41]
	Heterologous prime-boost vaccination with poxviruses expressing CEA, MUC-1, and costimulatory molecules with GM-CSF adjuvant	Increased OS in those with immune responses to CEA and/or MUC-1	[42]
	Carboplatin and paclitaxel ± oncolytic reovirus	No PFS improvement with addition of vaccine	[43]

**Table 2 cancers-17-02356-t002:** Selected ongoing clinical trials in pancreatic cancer with vaccine approaches.

Resectable/ Borderline Resectable			
	Trial No./Phase	Summary	Enrollment (Estimated)
Personal Neoantigen	NCT03558945 Phase 1	Distinct peptides grouped into 2–4 pools with poly-ICLC adjuvant given after surgery and adjuvant chemotherapy; sponsor: Anda Biopharmaceutical Development (Shenzhen) Co., Ltd.	30
	NCT05111353 Phase 1	Synthetic long peptide with poly-ICLC adjuvant administered after neoadjuvant chemotherapy and surgery or after neoadjuvant chemotherapy in the window before surgery; sponsor: Washington University School of Medicine.	34
	NCT04810910 Phase 1	Peptides with GM-CSF adjuvant given after surgical resection and adjuvant chemotherapy; sponsor: Zhejiang Provincial People’s Hospital.	20
	NCT06156267 Phase 1	mRNA vaccine administered after surgical resection and in combination with adebrelimab (anti-PD-L1) and mFOLFIRINOX; sponsor: Fudan University.	30
	NCT06496373 Phase 1	mRNA vaccine given after surgical resection and combined with a anti-PD-1; sponsor: Ruijin Hospital.	40
	NCT06326736 Phase 1	mRNA vaccine administered after surgical resection and camrelizumab (anti-PD-1), followed by adjuvant gemcitabine and Abraxane; sponsor: Jinling Hospital, China.	12
	NCT04161755 Phase 1	mRNA vaccine given after surgery and atezolizumab (anti-PD-L1), followed by adjuvant mFOLFIRINOX; sponsor: Memorial Sloan Kettering Cancer Center.	29
	NCT06353646	mRNA vaccine administered after surgery with ipilimumab (anti-CTLA-4) and adjuvant gemcitabine and capecitabine; sponsor: Wu Wenming.	12
	NCT04627246 Phase 1	Dendritic cell vaccine given after surgical resection combined with adjuvant chemotherapy and followed by nivolumab (PD-1 blockade); sponsor: Centre Hospitalier Universitaire Vaudois.	12
	NCT03552718 Phase 1	Yeast-based vaccine administered after completion of all curative therapy; sponsor: NantBioScience, Inc.	16
Shared Neoantigen	NCT05638698 Phase 2	RAS peptide vaccine (TG01) (with QS 21 adjuvant) with or without balstilimab (anti-PD-1) in patients with positive circulating tumor DNA (ctDNA) after surgical resection and adjuvant chemotherapy; sponsor: University of Kansas Medical Center.	24
	NCT04117087 Phase 1	Pooled mutant KRAS long-peptide vaccine with poly-ICLC adjuvant combined with nivolumab (anti-PD-1) and ipilimumab (anti-CTLA-4) after resection and adjuvant therapy; sponsor: Sidney Kimmel Comprehensive Cancer, Johns Hopkins.	30
	NCT02451982 Phase 2	GVAX vaccine in combination with cyclophosphamide, nivolumab (anti-PD-1), and urelumab (CD137 agonist) given prior to and after surgical resection (multiple combination arms); sponsor: Sidney Kimmel Comprehensive Cancer, Johns Hopkins.	76
	NCT05846516 Phase 1	ATP150/ATP152 (peptide-based prime vaccine) and VSV-GP154 (viral-based booster-vaccine) with ezabenlimab (anti-PD-1) administered after surgery and at least 3 months of peri-/adjuvant chemotherapy.	85
	NCT01088789 Phase 2	PANC 10.05 pcDNA-1/GM-Neo and PANC 6.03 pcDNA-1 neo vaccine (GM-CSF-secreting whole-tumor-cell vaccine) with cyclophosphamide administered after surgical resection and planned adjuvant therapy; sponsor: Sidney Kimmel Comprehensive Cancer, Johns Hopkins.	71
**Locally Advanced, Unresectable, or Metastatic Pancreatic Cancer**			
	**Trial No.**	**Summary**	**Enrollment**
Personal Neoantigen	NCT02600949 Phase 1	Peptide with imiquimod cream, pembrolizumab (anti-PD-1), and sotigalimab (CD40 agonist) in advanced pancreatic cancer; sponsor: M.D. Anderson Cancer Center.	150
	NCT05916261 Phase 1	mRNA vaccine with pembrolizumab (anti-PD-1) in advanced pancreatic cancer; sponsor: Ruijin Hospital.	54
Shared Neoantigen	NCT06577532 Phase 1	KRAS mRNA vaccine with or without toripalimab (anti-PD-1) in advanced pancreatic cancer; sponsor: Ruijin Hospital.	56
	NCT05964361 Phase 1/2	WT1-targeted dendritic cell vaccine transpresenting IL-15 in advanced pancreatic cancer; sponsor: University Hospital, Antwerp.	10
	NCT05721846 Phase 1	TGFβ-15 peptide vaccine with Montanide ISA-51 adjuvant combined with nivolumab (anti-PD-1) and ipilimumab (anti-CTLA-4) and stereotactic body radiotherapy (SBRT) in refractory pancreatic cancer; sponsor: Herlev Hospital.	20
	NCT06411691 Phase 1	Pooled mutant KRAS-targeted long peptide vaccine with balstilimab (anti-PD-1) and botensilimab (anti-CTLA-4) in unresectable or metastatic pancreatic cancer; Sidney Kimmel Comprehensive Cancer Center at Johns Hopkins.	50
	NCT00669734 Phase 1	Intratumoral CEA, MUC1 targeted viral vector with GM-CSF adjuvant in unresectable or metastatic pancreatic cancer; sponsor: National Cancer Institute (NCI).	18
	NCT06015724 Phase 2	KRAS-targeted vaccine with daratumumab (CD38 monoclonal antibody) and nivolumab (anti-PD-1) in advanced pancreatic cancer; sponsor: Georgetown University.	54
	NCT02432963 Phase 1	p53MVA viral vector with pembrolizumab (anti-PD-1) in advanced pancreatic cancer with TP53 overexpression; sponsor: City of Hope Medical Center.	11
**Preventative**			
	**Trial No.**	**Summary**	**Enrollment**
	NCT05013216 Phase 1	Mutant KRAS-targeted long-peptide vaccine with poly-ICLC adjuvant in patients with high risk of developing pancreatic cancer; sponsor: Sidney Kimmel Comprehensive Cancer Center at Johns Hopkins.	37

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
