# Peer review of "Cancer Vaccination and Immune-Based Approaches in Pancreatic Cancer"

_cancers, 2025, doi:10.3390/cancers17142356_

Round 1

Reviewer 1 Report

Comments and Suggestions for Authors

1) The abstract should be revised. The current abstract primarily presents background information and general facts, including the authors and the most important results (preferably numerical results). 

2)  I recommend that the authors draw a graphical abstract at the end of the introduction. In the graphical abstract, the authors should present the types of the vaccine, mechanisms of action, etc.  The authors did not use any figures at all, which is not good. 

3) Is it possible to develop vaccines against epigenetic markers? Such as non-coding RNAs (especially miRNAs) or DNA hypermethylation? 

4) In line 878, the authors wrote about exosome-based vaccines. It is good to describe more details about it in a new sub-section.  E.g., are there any exosome-based vaccines under clinical trials? 

Once again, this is a good manuscript, but the authors are required to add a graphical abstract with type and major characteristics of vaccines against PDAC, and draw a figure for each type of vaccine (containing mechanisms of action and detailed characteristics) 

Reviewer 2 Report

Comments and Suggestions for Authors

Dear Authors

You have made a nice review...a little be too long...but nice.

Your sentences are highlighted, my observations are not.

Pancreatic ductal adenocarcinoma (PDAC) is among the most common and lethal malignancies.

PDAC is the most lethal but not the most common cancer. It is the 7th more common cancer. Please correct the concept.

however, most patients are ineligible due to advanced disease at diagnosis

Please add that: 80 to 85% of patients are ineligible for surgery, and this rate has somewhat decreased with neoadjuvant treatments.

For those who undergo resection, adjuvant chemotherapy with a 6-month regimen of gemcitabine [2], gemcitabine plus capecitabine [3], or FOLFIRINOX (fluorouracil, leucovorin, irinotecan, and oxaliplatin) [4] has been shown to significantly improve outcomes

Add the gemcitabine, nab-paclitaxel association which is more frequentely used that the gemcitabine, capecitabine.

The choice of antigen(s), delivery platform, and clinical context (primary prevention, preoperative (neoadjuvant), after resection (adjuvant), or therapeutic for advanced disease) is variable [6].

I do not think vaccines are used for primary prevention of pancreatic cancer.

Between lines 100 and 120 a figure showing the immune mechanism would be highly convenient. Otherwise, delete all the textbook basic immunology explanations which are unnecessary for the experienced reader.

Some of the major players commonly involved in the TME include endothelial cells, cancer-associated fibroblasts (CAFs), and tumor-associated macrophages (TAMs) [17].

Here you omitted stellate cells, the most important component of the PDAC TME and which is different from CAFs. CAFs are fibroblasts, stellate cells are myofibroblasts. They have the same origin, that is mesenchymatic cells but CAFs and stellates cells morphology and effects are different.

The lack of mention of the immune functions of stellate cells calls my attention. See

Fu, Y., Liu, S., Zeng, S., & Shen, H. (2018). The critical roles of activated stellate cells-mediated paracrine signaling, metabolism and onco-immunology in pancreatic ductal adenocarcinoma. Molecular cancer17, 1-14.

Huang, Q., Huang, M., Meng, F., & Sun, R. (2019). Activated pancreatic stellate cells inhibit NK cell function in the human pancreatic cancer microenvironment. Cellular & Molecular Immunology16(1), 87-89.

Li, C., Cui, L., Yang, L., Wang, B., Zhuo, Y., Zhang, L., ... & Zhang, S. (2020). Pancreatic stellate cells promote tumor progression by promoting an immunosuppressive microenvironment in murine models of pancreatic cancer. Pancreas49(1), 120-127.

Li, H., Liu, D., Li, K., Wang, Y., Zhang, G., Qi, L., & Xie, K. (2024). Pancreatic stellate cells and the interleukin family: Linking fibrosis and immunity to pancreatic ductal adenocarcinoma. Molecular Medicine Reports30(3), 159.

Ene–Obong, A., Clear, A. J., Watt, J., Wang, J., Fatah, R., Riches, J. C., ... & Kocher, H. M. (2013). Activated pancreatic stellate cells sequester CD8+ T cells to reduce their infiltration of the juxtatumoral compartment of pancreatic ductal adenocarcinoma. Gastroenterology145(5), 1121-1132.

Extensive preclinical tumor models have demonstrated the strong effects of MDSCs in masking the antitumor efficacy of immune checkpoint blockade and chemotherapies.

This phrase requires a reference.

Around line 277 it would be convenient to incorporate a Table with the principal epitopes found in PDAC and determine if there are antibodies for these epitopes.

Moreover, driver KRAS mutations (mKRAS) present in >95% of PDAC.

I think that this proportion is slightly exaggerated. 85% is more realistic.

In the neoadjuvant setting, vaccines have been explored to improve conversion rates to surgery.

Here, you should also mention that immune checkpoint inhibitors (associated to chemotherapeutic drugs) in the neoadjuvant setting have decreased tumor relapse and improved overall survival.

In pancreatic cancer, however, their use has been limited to patients with microsatellite instability-high tumors (resulting in high neoantigen burdens), and even in this group, they have produced rare objective responses.

This phrase requires a reference.

Additionally, in patients treated with ipilimumab, a higher degree of clonal expansion was associated with longer OS.

I assume you are saying here:

Additionally, in patients treated with ipilimumab, a higher degree of T-cell clonal expansion was associated with longer OS.

One omitted issue is the “quasi” vaccination effect of Antibody Drug Conjugates (ADC) that release many epitopes while destroying the tumor cells. See:

Müller, P., Martin, K., Theurich, S., Schreiner, J., Savic, S., Terszowski, G., ... & Zippelius, A. (2014). Microtubule-depolymerizing agents used in antibody–drug conjugates induce antitumor immunity by stimulation of dendritic cells. Cancer immunology research2(8), 741-755.

Reviewer 3 Report

Comments and Suggestions for Authors

For section 2.1 on TCR activation, it should be emphasized which T cell populations require signals 1,2, and 3, since this is not true for all T cell subsets – especially in inflammatory environments where all bets are off. PMID:40425012 may be relevant. I'm just saying the paragraph makes it all feel outdated with this type of linear description/model. 

Section 2.2.1 was a little flaccid. Please be more specific - e.g. CAFs and TAMs are major players, but give an example of why to keep it a little more interesting and make me want to continue reading. 

What are N2 neutrophils?

Table 1 needs extensive reformatting. Why does any table need to be 4 pages long? Please add descriptive column names and reformat to generate a table, not a partitioned section of text. E.g. change "Methodology" to Stage/Methodology and consider adding more columns to remove repeating text/description.

Section on Tregs is underwhelming. If mechanisms underlying Treg accumulation and function have been extensively studied, please cite relevant literature. This cherry picking is poorly done - rewrite and add better references. 

TAM - this section should be improved by using more specific and correct statements. For example, "TAMs can reversibly alter their phenotype—termed polarization". Macrophage polarization does not refer to this cell type reversibly altering their phenotype, but rather refers to macrophages adopting a transcriptional/functional state. The way the text is written is confusing and misleading. Minor detail, but these theme keeps popping up throughout the text so please remove.

Neutrophils - what I just mentioned happens again: "TANs also exhibit strong plasticity—the ability to retain both N1 and N2 phenotype populations". No, plasticity does not refer to the ability to retain populations, but rather to the ability of a cell to change its phenotype or identity.

"Neutrophils contribute to immune suppression by secreting cytokines and chemokines and by inhibiting the activity of cytotoxic CD8⁺ T lymphocytes" – which cytokines? How is CD8 inhibition achieved? 

The lack of details in the neutrophils section is highlighted by the part that follows, where all of a sudden STAT3/Pax5 - BCL6 molecular pathways are discussed in the context of B cells. Saying this since it might be helpful to the reader to maintain consistency in the depth.

Table 2 needs reformatting similar to Table 1.

It is confusing that Tumor Infiltrating Lymphocytes is listed as an immunotherapy approach, since this refers to lymphocytes in the tumor in general which are discussed earlier in the review. 

Overall this review has a lot to offer after some editing. The fact that it is 35 pages long (albeit almost 10 of those taken up by tables) is a little over the top. This could be written in a more efficient way to not loose readers along the way. The tables should be edited and the text be written in a more specific to-the-point manner.

Great job guys, the next version will be great I hope. 

Round 2

Reviewer 1 Report

Comments and Suggestions for Authors

The authors provided the revision, and it is okay to publish.